# Successful kinetic impact into an asteroid for planetary defence

R. Terik Daly[1,28] ✉, Carolyn M. Ernst[1,28], Olivier S. Barnouin[1,28], Nancy L. Chabot[1], Andrew S. Rivkin[1], Andrew F. Cheng[1], Elena Y. Adams[1], Harrison F. Agrusa[2], Elisabeth D. Abel[1], Amy L. Alford[1], Erik I. Asphaug[3], Justin A. Atchison[1], Andrew R. Badger[1], Paul Baki[4], Ronald-L. Ballouz[1], Dmitriy L. Bekker[1], Julie Bellerose[5], Shyam Bhaskaran[5], Bonnie J. Buratti[5], Saverio Cambioni[6], Michelle H. Chen[1], Steven R. Chesley[5], George Chiu[1], Gareth S. Collins[7], Matthew W. Cox[1], Mallory E. DeCoster[1], Peter S. Ericksen[1], Raymond C. Espiritu[1], Alan S. Faber[1], Tony L. Farnham[2], Fabio Ferrari[8], Zachary J. Fletcher[1], Robert W. Gaskell[9], Dawn M. Graninger[1], Musad A. Haque[1], Patricia A. Harrington-Duff[1], Sarah Hefter[1], Isabel Herreros[10], Masatoshi Hirabayashi[11], Philip M. Huang[1], Syau-Yun W. Hsieh[1], Seth A. Jacobson[12], Stephen N. Jenkins[1], Mark A. Jensenius[1], Jeremy W. John[1], Martin Jutzi[13], Tomas Kohout[14,15], Timothy O. Krueger[1], Frank E. Laipert[5,27], Norberto R. Lopez[1], Robert Luther[16], Alice Lucchetti[17], Declan M. Mages[5], Simone Marchi[18], Anna C. Martin[1], Maria E. McQuaide[1], Patrick Michel[19], Nicholas A. Moskovitz[20], Ian W. Murphy[1], Naomi Murdoch[21], Shantanu P. Naidu[5], Hari Nair[1], Michael C. Nolan[3], Jens Ormö[10], Maurizio Pajola[17], Eric E. Palmer[9], James M. Peachey[1], Petr Pravec[22], Sabina D. Raducan[13], K. T. Ramesh[23], Joshua R. Ramirez[1], Edward L. Reynolds[1], Joshua E. Richman[1], Colas Q. Robin[21], Luis M. Rodriguez[1], Lew M. Roufberg[1], Brian P. Rush[5], Carolyn A. Sawyer[1], Daniel J. Scheeres[24], Petr Scheirich[22], Stephen R. Schwartz[9], Matthew P. Shannon[1], Brett N. Shapiro[1], Caitlin E. Shearer[1], Evan J. Smith[1], R. Joshua Steele[1], Jordan K. Steckloff[9], Angela M. Stickle[1], Jessica M. Sunshine[2], Emil A. Superfin[1], Zahi B. Tarzi[5], Cristina A. Thomas[25], Justin R. Thomas[1], Josep M. Trigo-Rodríguez[26], B. Teresa Tropf[1], Andrew T. Vaughan[5], Dianna Velez[5], C. Dany Waller[1], Daniel S. Wilson[1], Kristin A. Wortman[1] & Yun Zhang[2]

Although no known asteroid poses a threat to Earth for at least the next century, the catalogue of near-Earth asteroids is incomplete for objects whose impacts would produce regional devastation[1,2]. Several approaches have been proposed to potentially prevent an asteroid impact with Earth by deflecting or disrupting an asteroid[1–3]. A test of kinetic impact technology was identified as the highest-priority space mission related to asteroid mitigation[1]. NASA's Double Asteroid Redirection Test (DART) mission is a full-scale test of kinetic impact technology. The mission's target asteroid was Dimorphos, the secondary member of the S-type binary near-Earth asteroid (65803) Didymos. This binary asteroid system was chosen to enable ground-based telescopes to quantify the asteroid deflection caused by the impact of the DART spacecraft[4]. Although past missions have utilized impactors to investigate the properties of small bodies[5,6], those earlier missions were not intended to deflect their targets and did not achieve measurable deflections. Here we report the DART spacecraft's autonomous kinetic impact into Dimorphos and reconstruct the impact event, including the timeline leading to impact, the location and nature of the DART impact site, and the size and shape of Dimorphos. The successful impact of the DART spacecraft with Dimorphos and the resulting change in the orbit of Dimorphos[7] demonstrates that kinetic impactor technology is a viable technique to potentially defend Earth if necessary.

The DART spacecraft was launched on 24 November 2021. The spacecraft carried a narrow-angle imager called the Didymos Reconnaissance and Asteroid Camera for Optical navigation (DRACO), which was used for optical navigation, terminal guidance and asteroid characterization[8]. DRACO detected Didymos—the primary asteroid in the binary system—in summed optical navigation images 61 days before impact.

On 27 August 2022, 30 days before impact, DRACO began taking optical navigation images of Didymos every 5 hours, which were processed by the ground optical navigation team[9].

On 26 September 2022 at 19:09:24 UTC, 4 h and 5 min before impact, the spacecraft's autonomous Small-body Maneuvering Autonomous Real Time Navigation (SMART Nav) system[10] took control of spacecraft

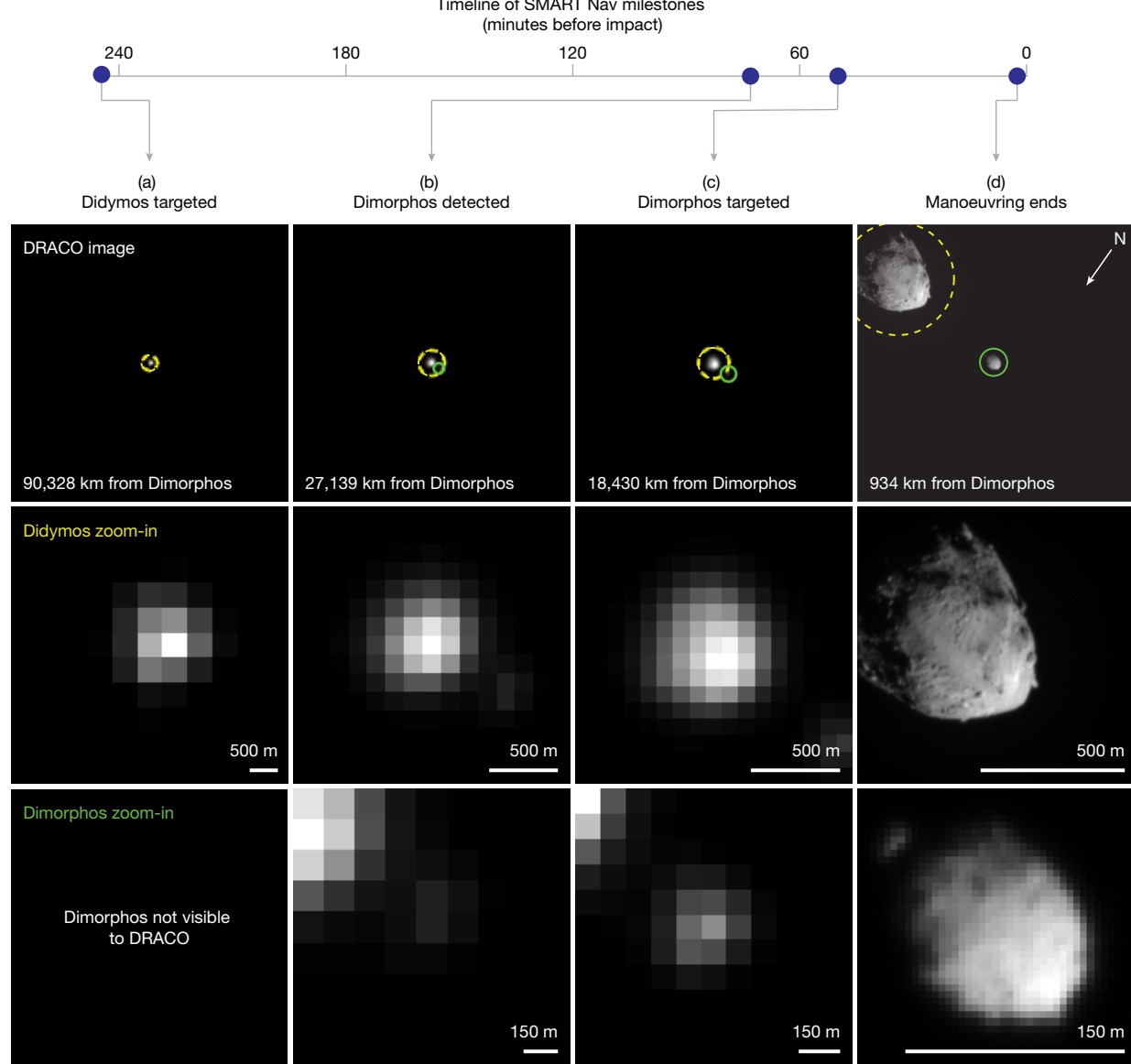

**Timeline of SMART Nav milestones**
(minutes before impact)

240   180   120   60   0

(a)
Didymos targeted

(b)
Dimorphos detected

(c)
Dimorphos targeted

(d)
Manoeuvring ends

DRACO image

90,328 km from Dimorphos | 27,139 km from Dimorphos | 18,430 km from Dimorphos | 934 km from Dimorphos

Didymos zoom-in

500 m | 500 m | 500 m | 500 m

Dimorphos zoom-in

Dimorphos not visible to DRACO

150 m | 150 m | 150 m

**Fig. 1 | Milestones leading to the impact with Dimorphos from the time SMART Nav began targeting until the end of SMART Nav manoeuvring.** **a**–**d**, Each column corresponds to a milestone: Didymos targeted (**a**), Dimorphos detected (**b**), Dimorphos targeted (**c**) and end of manoeuvring (**d**). Each row shows, from top to bottom, the raw DRACO image at the time of that milestone with circles indicating the two asteroids detected by onboard processing and identified by SMART Nav (yellow dashed circles, Didymos; green solid circles, Dimorphos), a zoom-in of Didymos and a zoom-in of Dimorphos. The SMART Nav system used information in DRACO images to successfully impact Dimorphos. In all images, the north pole of Dimorphos (+Z) is towards the bottom left. Images from left to right: dart_0401915351_36903_01_raw.fits, dart_0401925635_06853_01_raw.fits, dart_0401927052_23729_01_raw.fits and dart_0401929899_33346_01_raw.fits.

navigation (Fig. 1a). SMART Nav processed DRACO images onboard[11] to identify Didymos, and once resolved, Dimorphos. To achieve impact with Dimorphos, the spacecraft needed to distinguish between the two asteroids and hit the smaller, dimmer one. Owing to the dynamics of the binary system and the spatial resolution of DRACO, it was known that the spacecraft's ultimate target—Dimorphos—would be hidden from DRACO's view during most of the autonomous phase. By design, SMART Nav manoeuvred the spacecraft towards Didymos until Dimorphos became reliably detected[10]. SMART Nav first detected Dimorphos 73 min before impact, and at 50 min before impact, SMART Nav began manoeuvring towards Dimorphos (Fig. 1). As planned, SMART Nav manoeuvring ended at 23:11:52 UTC, 2.5 min before impact, to give the spacecraft time to settle to minimize jitter and smear in the final images. The spacecraft impacted Dimorphos at 23:14:24.183 ± 0.004 UTC (Methods). From the start of

autonomous navigation until impact, the spacecraft continuously streamed images to the ground, which were immediately broadcast to the public. The final full image was acquired 1.818 s before impact and has a pixel scale of 5.5 cm. The final image received on the ground was a partial image acquired 0.855 s before impact with a pixel scale of 2.6 cm.

Little was known about the shape or surface of Dimorphos until DRACO obtained high-resolution images. Ground-based radar observations[12] led to a diameter estimate of 150 ± 30 m. Analysis of telescopic photometric observations yielded a comparable diameter for Dimorphos, 171 ± 11 m (refs. 13,14). Although DRACO imaged only a portion of Dimorphos, and illumination was limited at 60° solar phase, the images were used to construct an asteroid shape model (Methods). The shape model revealed Dimorphos to be an oblate spheroid with a volume-equivalent diameter of 151 ± 5 m (Table 1, Fig. 2 and Extended Data Fig. 1). The shape

## Table 1 | Properties of the DART impact, Dimorphos and Didymos

| | |
|---|---|
| Time of impact | 26 September 2022 at 23:14:24.183±0.004 UTC |
| Impact speed (km s⁻¹) | 6.1449±0.0003 |
| Impact angle (°) | 73±7 from local horizontal, 17±7 from the surface normal |
| Impact site location (latitude, longitude) | 8.84±0.45° S, 264.30±0.47° E |
| Impact site offset from centre of figure (m) | 25±1 |
| Spacecraft mass at time of impact (kg) | 579.4±0.7 |
| Impact kinetic energy (GJ) | 10.94±0.01 |
| Extent of Dimorphos (m) | $X$: 177±2<br>$Y$: 174±4<br>$Z$: 116±2 |
| Extent of Didymos (m) | $X$: 849±15<br>$Y$: 851±15<br>$Z$: 620±15 |
| Volume of Dimorphos (km³) | 0.00181±10% |
| Diameter of volume-equivalent sphere for Dimorphos (m)ᵃ | 151±5 |
| Diameter of volume-equivalent sphere for Didymos (m)ᵃ | 761±26 |
| Mass of the Didymos system (kg) | (5.6±0.5)×10¹¹ |
| Density of Didymos system (kg m⁻³) | 2,400±300 |
| Inferred mass of Dimorphos (kg) | 4.3×10⁹ |

ᵃComputed from the volumes of the shape models, not the volumes of triaxial ellipsoids with the extents listed above.

of Dimorphos is unusual relative to other near-Earth asteroids visited by spacecraft[15–19] and differs from other binary asteroid secondaries observed so far that have measured elongations[20–23]. However, oblate secondaries show little or no measurable light-curve amplitude, which biases the observational sample towards elongated secondaries. A size estimate for Didymos from DRACO images (Table 1 and Methods) combined with previous telescopic observations[4] enables calculation

of a more accurate visible (0.55 μm) geometric albedo for the system of 0.15 ± 0.02. This value is on the low side, but within 1σ, of the mean geometric albedo for S-type asteroids[24].

The spacecraft trajectory and pointing were reconstructed to locate the impact site (Methods and Fig. 2). The spacecraft impacted Dimorphos at 8.84 ± 0.45° S, 264.30 ± 0.47° E, within 25 m of the centre of the figure, which is very near the scenario for maximizing momentum transfer with an impact through the centre of the figure[25]. The 1σ uncertainty in the impact site location is ±68 cm (Methods), which is smaller than the size of the spacecraft bus (Fig. 3). The impact angle was 73 ± 7° from local horizontal (Methods and Extended Data Fig. 2). The impact site was near two large boulders, labelled boulder 1 (6.5-m long and at its highest point standing about 2.2 m above the surrounding terrain; Extended Data Fig. 3) and boulder 2 (6.1-m long and at its highest point standing about 1.6 m above the surrounding terrain; Extended Data Fig. 3) in Fig. 2b,c. The spacecraft approached the asteroid with its solar arrays canted slightly towards the surface (Fig. 3). The leading edge of the +Y solar array contacted the surface of boulder 1, and this solar array directly hit boulder 1 (Fig. 3). Almost immediately thereafter, the −Y solar array grazed boulder 2, with the leading edge of the −Y array contacting the surface near the base of boulder 2 in downrange direction (Fig. 3). Finally, the spacecraft bus hit between boulders 1 and 2 (Fig. 3). Although the solar arrays contacted Dimorphos just before the spacecraft bus, the bulk of the spacecraft's energy was transferred by the bus, which accounted for about 88% of the spacecraft mass at the time of impact.

DART images of Dimorphos revealed a boulder-strewn surface (Fig. 2a, and Extended Data Figs. 4 and 5) resembling other small near-Earth asteroids, such as the S-type (25143) Itokawa[17], and carbonaceous asteroids (101955) Bennu[26] and (162173) Ryugu[16], suggesting a rubble-pile structure for Dimorphos. The boulder-rich nature of the surface is apparent in images as coarse as 2–3-m pixel scale (Extended Data Fig. 4). No unambiguous impact crater candidates are observed, which indicates a young surface, although craters can be difficult to identify on boulder-covered terrains[27–30]. The appearance of Dimorphos contrasts with impressions from lower-resolution images of the Didymos surface, where regional roughness variations are apparent (Extended Data Fig. 4).

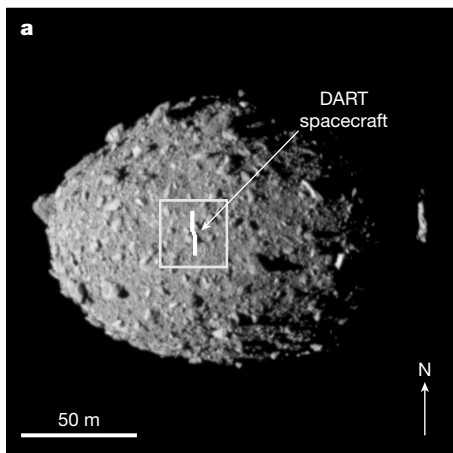
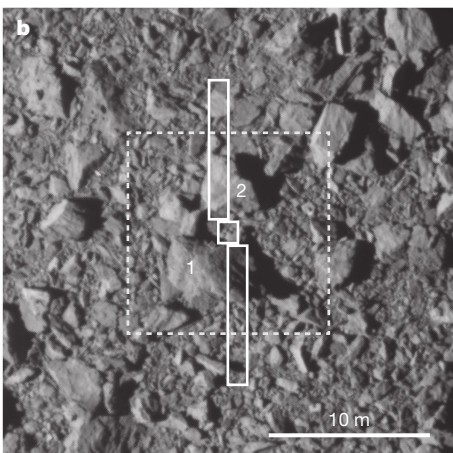
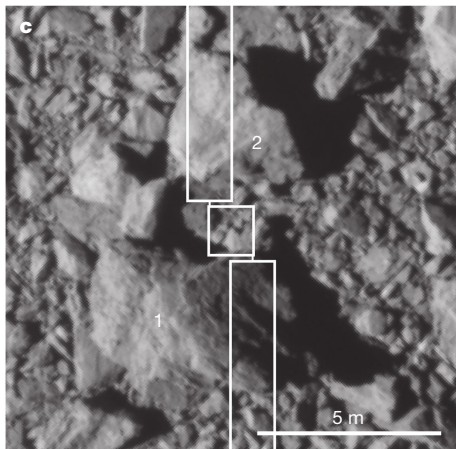

**Fig. 2 | The asteroid Dimorphos and the DART impact site as seen in calibrated DRACO images. a**, Dimorphos with an appropriately scaled and correctly oriented outline of the DART spacecraft centred on the impact site. Note the size of the spacecraft relative to the asteroid. The spacecraft bus was approximately 1.2 × 1.3 × 1.3 m, from which other structures extended, resulting in dimensions of approximately 1.8 × 1.9 × 2.6 m. The spacecraft also had two large solar arrays that were each 8.5 m long. **b**, A closer view of the DART impact site showing the outline of the spacecraft bus and solar arrays over the DRACO image. Note the positions of the two solar arrays relative to two large boulders,

labelled 1 (6.5 m long) and 2 (6.1 m long). This subframe is from an image taken 2.781 s before impact. **c**, The spacecraft bus hit between boulders 1 and 2, whereas the solar arrays interacted with these boulders. This subframe is from an image taken 1.818 s before impact. The arrow in the bottom right of **a** indicates the direction of the Dimorphos +Z (north) axis. The solid white box in **a** shows the location of the image in **b**. The dashed white box in **b** shows the location of the image in **c**. Panels **b** and **c** show subimages of the full frame. Image names: dart_0401930039_14119_01_iof.fits (**a**), dart_0401930048_45552_01_iof.fits (**b**) and dart_0401930049_43695_01_iof.fits (**c**).

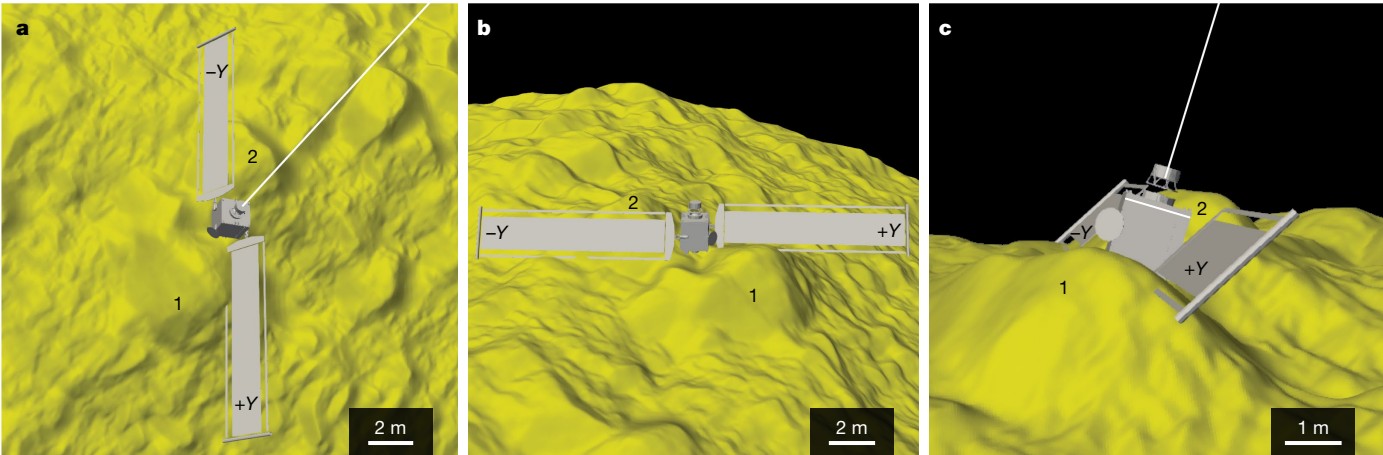

**Fig. 3 | Relationship between the spacecraft and topography at the DART impact site. a**–**c**, The position of the spacecraft immediately before the impact of the spacecraft bus from different perspectives to visualize the three-dimensional interactions between the spacecraft and surface. **a**, Dimorphos north is towards the top of the panel. **b**, Dimorphos north is to the right. **c**, Dimorphos north is roughly into the page. In all panels, the −*Y* solar array points to Dimorphos north. Length scales vary in these perspective views; the scale bars shown are approximate. Boulders 1 and 2 correspond to boulders 1 and 2 in Fig. 2. The caption to Fig. 2 gives the spacecraft dimensions.

The final complete DRACO image captured an approximately 880-m² area surrounding the impact site at a 5.5-cm pixel scale. The impact region (Fig. 2b) exhibits blocky terrain resembling the rest of the observed hemisphere (Fig. 2a, and Extended Data Figs. 4 and 5). There is evidence for variation within boulders, 'rocks on rocks' similar to observations on Bennu[31], and partially buried boulders (Extended Data Fig. 5). The longest axes of boulders counted in the final complete image are 0.16 m to 6.5 m in length. The impact region has fewer boulders in the 0.2–0.5 m size range than expected if the cumulative distribution followed a single power law, even though the pixel scale of the image is sufficient for their identification (Extended Data Fig. 6). There is no evidence for expansive smooth deposits (grain size smaller than the image pixel scale) such as those seen on Itokawa[17]. The blocky nature of the impact site probably influenced crater formation, ejecta and momentum enhancement, as seen in impact experiments[28,32–34], numerical simulations[25,35] and the Small Carry-on Impactor experiment on Hayabusa2[5].

DART did not measure the mass of Dimorphos. Instead, the mass of Dimorphos is estimated using the orbital properties of the binary system, the total volume of the system and an assumption that Didymos and Dimorphos have equal bulk densities (Methods). This assumption cannot be rigorously tested from DART data, but this approach leads to a bulk density of Dimorphos of 2,400 kg m⁻³ (Table 1) with difficult-to-quantify uncertainties.

On the basis of analyses of reflectance spectra, the best meteoritic analogues for Didymos are L and LL chondrites[36,37]. L and LL chondrites have grain densities[38] of 3,580 ± 10 kg m⁻³ and 3,520 ± 10 kg m⁻³, respectively. If one assumes Dimorphos has the same composition as Didymos and that these meteorite values represent the grain density of Dimorphos, then a bulk density of about 2,400 kg m⁻³ implies a Dimorphos bulk porosity of the order of 30% (with a difficult-to-quantify uncertainty; Methods). This level of bulk porosity is not inconsistent with a rubble-pile structure for Dimorphos, a structure suggested by the boulder-rich character of Dimorphos's surface. This bulk porosity probably exists as a combination of macroporosity between pieces of rubble and microporosity within individual pieces of rubble. L and LL chondrite samples have porosities of 8.0 ± 0.3% and 9.5 ± 0.6%, respectively[38], which would imply that macroporosity is substantial on Dimorphos. Estimates of the density and porosity of Dimorphos will improve when the European Space Agency's Hera mission arrives at the Didymos system in early 2027[39].

DART's successful autonomous targeting of a small asteroid with limited prior knowledge is a key accomplishment on the path to advancing kinetic impactor technology to an operational capability. The impact of DART indicates that a precursor reconnaissance mission is not a prerequisite for intercepting a subkilometre asteroid, although the characterization done by a precursor mission would provide valuable information for optimizing, planning and predicting the outcome with greater certainty. Kinetic impactor technology for asteroid deflection requires having sufficient warning time—at least several years but preferably decades—to prevent an asteroid impact with Earth[1–3]. Nevertheless, this successful step to demonstrate the viability of kinetic impactor technology for planetary defence builds optimism about humanity's capacity to protect Earth from an asteroid threat.

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

¹Johns Hopkins University Applied Physics Laboratory, Laurel, MD, USA. ²University of Maryland, College Park, MD, USA. ³University of Arizona, Tucson, AZ, USA. ⁴Technical University of Kenya, Nairobi, Kenya. ⁵Jet Propulsion Laboratory, California Institute of Technology, Pasadena, CA, USA. ⁶Massachusetts Institute of Technology, Cambridge, MA, USA. ⁷Imperial College London, London, UK. ⁸Politecnico di Milano, Milan, Italy. ⁹Planetary Science Institute, Tucson, AZ, USA. ¹⁰Centro de Astrobiologiá (CAB) CSIC-INTA, Torrejón de Ardoz, Spain. ¹¹Auburn University, Auburn, AL, USA. ¹²Michigan State University, East Lansing, MI, USA. ¹³University of Bern, Bern, Switzerland. ¹⁴Institute of Geology of the Czech Academy of Sciences, Prague, Czech Republic. ¹⁵University of Helsinki, Helsinki, Finland. ¹⁶Museum für Naturkunde, Leibniz Institute for Evolution and Biodiversity Science, Berlin, Germany. ¹⁷INAF-Astronomical Observatory of Padova, Padua, Italy. ¹⁸Southwest Research Institute, Boulder, CO, USA. ¹⁹Université Côte d'Azur, Observatoire de la Côte d'Azur, CNRS, Laboratoire Lagrange, Nice, France. ²⁰Lowell Observatory, Flagstaff, AZ, USA. ²¹ISAE-SUPAERO, Université de Toulouse, Toulouse, France. ²²Astronomical Institute AS CR, Ondrejov, Czech Republic. ²³Johns Hopkins University, Baltimore, MD, USA. ²⁴University of Colorado, Boulder, CO, USA. ²⁵Northern Arizona University, Flagstaff, AZ, USA. ²⁶Institute of Space Sciences, Barcelona, Spain. ²⁷Present address: Nabla Zero Labs, South Pasadena, CA, USA. ²⁸These authors contributed equally: R. Terik Daly, Carolyn M. Ernst, Olivier S. Barnouin. ✉e-mail: terik.daly@jhuapl.edu

## Methods

### Determining the time of impact

The time of impact was computed from spacecraft clock timestamps inserted into each downlinked telemetry frame by the spacecraft's radio. A new telemetry frame was produced every 2.9853 ms with a precision of 20 µs. Assuming that the impact occurred in the middle of the frame, the final received frame placed the time of impact to within ±half a frame period: 401,930,051.59326 ± 0.00149 s spacecraft clock time. This spacecraft clock time of impact was converted to UTC. This conversion increases the uncertainty slightly: 2022-26-9 23:14:24.183 ± 0.004 s UTC. This time is the UTC of the impact at Dimorphos, not the time on Earth when the last telemetry frame was received. As each DRACO image is timestamped with a spacecraft clock value, the time to impact for each image can be computed with similar accuracy.

### Shape modelling of Dimorphos

We built the shape model using stereophotoclinometry (SPC), a technique that has been widely used to model the shapes of small bodies[40–42]. Before the impact of DART, the shape modelling team conducted tests to understand the strengths and limitations of the SPC approach given the data expected from DART[43]. DART images pose a challenge for any image-based shape modelling technique owing to the single viewing geometry and lighting condition. SPC provides robust results despite these challenges[43].

Extended Data Fig. 7 illustrates the steps used to build the Dimorphos shape model. First, we used DRACO images to determine the dimensions of an initial triaxial ellipsoid. The ellipsoid was constrained by the location of the sunlit limb, the location of the limb lit by light scattered off Didymos and the position of the terminator. Together, the sunlit and Didymos-lit limbs revealed the complete extents of the $X$ and $Z$ axes of Dimorphos (Extended Data Fig. 8). The position of the terminator constrained the $Y$-axis extent. The last DRACO image to contain all of Dimorphos in the field of view was registered to the ellipsoid. Other DRACO images were registered to that image or, in the case of the highest-resolution images of the impact site, to the immediately preceding image. To correctly determine the scale of the shape model, we used the known time of impact and spacecraft speed to set the spacecraft range for each image.

Next, we pursued parallel paths (Extended Data Fig. 7). The first path used standard SPC processes to construct small digital terrain models (called 'maplets') using an SPC program called lithos[40,42,43]. Maplets were tiled all across the surface of the asteroid in the areas seen by DRACO. Maplets went through several iterations to compute the maplet topography[40,42,43]. After several iterations, the maplet ensemble was averaged to create a global digital terrain model (DTM). Areas without maplet coverage remained unchanged. This cycle was repeated with maplets of increasingly finer ground-sample distances (GSDs) in the areas covered by higher-resolution images surrounding the impact site. The finest-scale maplets had a GSD of 3 cm, comparable to the pixel scale of the final, partial DRACO image. The second path focused on matching the sunlit limb. We built maplets along the sunlit limb that were conditioned by limb points identified in the tilt-to-height integration in an SPC iteration. These maplets were made at only two GSDs owing to the coarser resolution of images that contained limb. Once the topography in the standard maplets and limb-only maplets stabilized, we united them via several SPC iterations and proceeded to build the global DTM.

The areas covered by maplets in the Dimorphos global DTM are shown in Extended Data Fig. 1 as shaded grey regions. The blue and magenta points show the locations of the sunlit and Didymos-lit limbs. The unseen side of the asteroid is roughly centred on the 90° E view. The shape model has the volume and extents reported in Table 1. The global DTM has a typical GSD of 26 cm, but that resolution is only meaningful in the areas covered by maplets.

We also used the standard maplets to construct a DTM of the impact site. The impact site is covered by the highest-resolution images, so the impact site can support finer-scale topography than the global model. The impact site DTM has a GSD of 5 cm and is shown in Fig. 3.

Owing to the short period of time for which we have resolved images of Dimorphos, the rotation pole, prime meridian and rotation rate of Dimorphos could not be updated using SPC. Instead, we used values derived by ground-based observers. In the equatorial J2000 frame, the pole values used for Dimorphos are[14]: BODY120065803_POLE_RA = (60.936309840897856, 0, 0) and BODY120065803_POLE_DEC = (−71.674565992873852, 0, 0).

The coordinate system of the global DTM of Dimorphos has the prime meridian pointed towards Didymos, consistent with International Astronomical Union convention: BODY120065803_PM = (64.914870 949788195, 724.723943017441570, 1.0840372309 × 10⁻⁶).

A planetary constants kernel (PCK) with these values is available at https://ssd.jpl.nasa.gov/ftp/eph/small_bodies/dart/dimorphos/archive/ called 'dimorphos_s501-preimp.tpc'. Documentation for PCK files, including units for the terms listed above, is available at https://naif.jpl.nasa.gov/pub/naif/toolkit_docs/FORTRAN/req/pck.html#Text%20PCK%20Kernel%20Variable%20Names. The north pole of Dimorphos is in the $+Z$ direction of the shape model but oriented towards ecliptic south.

### Shape model uncertainties

We quantified uncertainties in the global DTM of Dimorphos using techniques employed to estimate the uncertainties associated with the global DTM of Bennu[44]. These techniques were also used to assess uncertainties in shape models developed during pre-impact tests done for DART[43]. The analyses involved comparing DRACO images against the global DTM when rendered with the same lighting and viewing geometry as the image. We used three analyses described in the above references. The first method, referred to as the limb and terminator method, used image thresholding to identify the limb and terminator in the DRACO image and rendered shape model. The thresholded images were subtracted to reveal mismatches between the image and rendered shape model and any size bias in the model (Extended Data Fig. 9a–e). The second and third methods relied on analysis of corresponding surface features, or keypoints, in images and the rendered shape model, to understand errors in the overall size of the shape model (Extended Data Fig. 9f–i). In the second method, referred to as keypoint matching, the rendered image was rotated, translated and scaled to minimize differences in keypoint locations on the DRACO image and rendered shape model. The third method, referred to as keypoint distance, was based on the measured distances between all keypoints in the DRACO image and rendered shape model.

The limb and terminator assessments indicated that the model is 75 cm too small in $X$, $Y$ and $Z$, with a limb uncertainty of 1.3–2 m. Most of the mismatch between DRACO images and the rendered shape model occurred along the limb and terminator. We expected larger uncertainties in these areas owing to the limited coverage and single lighting condition in DRACO images. The keypoints indicated that the Dimorphos shape model is 6 ± 29 cm too large and that features in the model have point-separation errors of −11 ± 20 cm. The keypoint assessments indicated that the model performed well in areas where DRACO resolved detailed surface features, which is the area of the model that is most relevant to understanding the topography of the impact site. Comparisons between shadow lengths in DRACO images and shadow lengths in shape model renderings indicated that some boulder heights are too small. This mismatch was expected based on shape modelling tests done for DART and is a consequence of constructing the shape model from a single lighting condition and viewing geometry[43]. Shadow lengths suggest that the height of boulder 1 in the impact site DTM is about 10% too small, but the

height of boulder 2 in the impact site DTM is correct. On the basis of all of these analyses, we assumed uncertainties of 2 m in the extents of Dimorphos in the $X$ and $Z$ directions.

Because DRACO saw a complete outline of the asteroid (Extended Data Fig. 8) and the spacecraft approach geometry was such that this outline was primarily in the $X$–$Z$ plane, the shape model dimension uncertainties in $X$ and $Z$ are less than 1% (Table 1). Hence, the $Y$-axis extent is the largest source of uncertainty in the volume. Future work will refine the errors in the $Y$ axis, but for the time being we assume that the uncertainty in $Y$ is two times the uncertainties in $X$ and $Z$ (that is, 4 m).

Measuring the volume of Dimorphos and realistically estimating uncertainties on that quantity is of particular interest because it directly affects our understanding of the mass of the asteroid. In the shape modelling tests[43], we took scaled shape models of an ellipsoid, Itokawa, and Bennu, rendered a set of simulated DRACO images from the 'truth' shape models, and then used SPC to build shape models from simulated images[43]. We compared the volumes of the models built using SPC against the volumes of truth models[43]. The volume errors in those tests ranged from −2% to +23% (ref. 43). The test with the lowest volume error was for an ellipsoidal asteroid. The roughly ellipsoidal shape of Dimorphos as seen by DRACO suggests that the volume error in the Dimorphos global DTM is probably on the order of several per cent, rather than 23%, by virtue of the simplicity of the long-wavelength shape of Dimorphos. Moreover, in the tests with the largest volume errors, the terminator of the shape model—which is the primary constraint on the extent of the shape into and out of the page—did not match the terminator of the original images. In the case of Dimorphos, the terminator matched nicely between the global DTM and the rendered shape model (Extended Data Fig. 9). Given the results from the shape modelling tests[43], this agreement indicated that the volume errors for Dimorphos are probably on the order of several per cent. On the basis of these assessments, a volume error of 5% seems probable. But, we adopted a volume uncertainty of 10% to be conservative. This volume error is larger than the value implied by the reported uncertainties in the $X$, $Y$ and $Z$ extents, but the intent is to be conservative.

## Shape modelling of Didymos

A preliminary SPC shape model of Didymos was built from DRACO data. This model led to the preliminary volume and extent of Didymos reported in Table 1. DRACO images revealed a smaller $Z$ axis and showed that the visible portion of the $X$ axis probably needs to be extended by a few tens of metres compared with a radar-derived shape model[12]. We used the techniques employed to estimate shape model uncertainties described in the 'Shape modelling of Dimorphos' section to assess the preliminary SPC shape model of Didymos. Because the full $X$ and $Z$ extents of Didymos cannot be measured in DRACO images (in contrast to Dimorphos), we conservatively attributed an uncertainty of 15 m to the lengths of all the axes of Didymos.

## Impact site identification

The construction of an SPC shape model produced a set of surface landmarks that were used to determine the location (at the time of each image) of the spacecraft relative to the surface in the Dimorphos body fixed frame. This procedure used the spacecraft's Didymos relative velocity computed by the ground navigation team using a combination of radiometric tracking data (Doppler, range) and optical images of Didymos. We used this information and the shape model of Dimorphos to compute the location of the impact site. The positions of the DART spacecraft from SPC were converted to the inertial J2000 frame and corrected for light time and aberration. The velocity of the spacecraft was estimated by fitting a second-order polynomial function to these positions as a function of time. This approach is identical to that employed previously[45] to estimate the position of the Hayabusa

spacecraft relative to Itokawa. We used the locations from the last 14 complete images collected by DART to determine the spacecraft velocity. These images contained large numbers of landmarks due to their fine pixel scales, which help anchor the spacecraft position for each image. The fit residuals (that is, the difference between the fit of the spacecraft location to that determined from SPC) are <1 m (Extended Data Fig. 10).

We combined our estimated velocity and the SPC-derived spacecraft positions of the last five images to determine the impact location. Only the last five images were used to locate the impact because heater cycling on the spacecraft introduced cyclic error rates in the inertial attitude knowledge. These rates were estimated to be small at the time of impact owing to the timing of impact relative to heater cycling but at their highest approximately 30 s before impact. Using only the last few images reduced the influence of this known (and cumulative) error source. At each of these five spacecraft positions, we propagated the velocity vector until it intersected with the surface of the Dimorphos shape model. We took the mean of these positions as the impact location. We also computed the spacecraft state at a few different heights above the intersection point to determine the order in which the solar arrays and bus contacted the surface. The propagation from each of the last five images provided the same impact point to within 1 cm. The uncertainties in the impact point location (reported in Table 1) are dominated by the residuals to the fits of the spacecraft position obtained by SPC in the $Y$ and $Z$ directions (Extended Data Fig. 10).

## Impact-angle assessments

The tilt angle of the surface relative to the impact velocity vector defines the impact angle. Because the impact site DTM resolves topography at a 5-cm GSD, the spacecraft bus would have interacted with on the order of a thousand facets of the DTM. Therefore, we calculated a mean tilt with respect to the impact velocity vector for each facet in the impact site DTM. The mean tilt calculation is based on previous work[41]; however, we computed mean tilt with respect to the impact velocity vector, $\mathbf{v}$, rather than the radial vector to a given facet to determine the impact angle.

The magnitude of the mean tilt for a facet at the impact point is the angle between $\mathbf{v}$ and the average normal, $\mathbf{n}_{av}$, which is the weighted average of the normal vectors of all facets in the user-defined region, $\mathbf{n}_{av} = \sum(\mathbf{n}_i A_i)/\sum A_i$ surrounding the impact point. In this study, this region was 1.5 m in radius to exceed the size of the spacecraft bus. The normal vector of each facet in the region of interest, $\mathbf{n}_i$, was weighted by $A_i$, the area of the facet projected onto a best-fit plane to the region selected to determine the surface tilt. This yields a mean tilt of acos $(\mathbf{n}_{av} \times \mathbf{v}/|\mathbf{n}_{av}||\mathbf{v}|)$. The impact angle, $\theta$, which is typically defined relative to the local horizontal for planetary impacts, is given by $\theta = 90 - $ (mean tilt). The impact angle shown in Table 1 is computed from the mean tilt and $\theta$ of the facet closest to the impact point.

As discussed in the 'Impact site identification' section, the location of the impact site has an uncertainty of ±68 cm. To understand the range of mean tilts that DART may have encountered given the uncertainty in the impact location, we considered the distribution of mean tilt for all facets within a circle with radius 68 cm centred on the impact site (Extended Data Fig. 2). The standard deviation of this distribution was 7° (1$\sigma$), which we attributed as the uncertainty of the impact angle DART may have experienced.

## Boulder counting

The longest axis of each boulder in the final full DRACO image was identified as a line (as done for asteroid (101955) Bennu[46]). The length of the longest axis was determined from the line length and image pixel scale, assuming the last full image as a 'flat' scene. The total number of boulders and pebbles identified in the final full DRACO image impact site is 953 and range in size (that is, the length of the longest axis) from 0.16 m (limit of image resolution, assuming ≥3-pixel sampling[47]) to

6.5 m. The resulting size–frequency distribution is shown in Extended Data Fig. 6.

## Estimates for the density of Dimorphos

To first order, the combined system mass, $M_{sys}$, was estimated using Kepler's third law

$$M_{sys} = (4\pi^2 a^3)/(GP^2)$$

where $a$ and $P$ are the pre-impact semimajor axis and the orbit period, respectively, and $G$ is the gravitational constant. Neglecting the aspherical shapes of Didymos and Dimorphos and their associated gravitational potentials may lead to an overestimate of the system mass by about 1–2% (ref. 48). However, this error is negligible as the uncertainty in Dimorphos's semimajor axis dominates the uncertainty in the system mass. Next, the bulk density of the combined system was obtained by dividing the system mass with the combined volume of both bodies

$$\rho_{sys} = (3M_{sys})/(4\pi[R_A^3 + R_B^3])$$

where $R_A$ and $R_B$ are the volume-equivalent radii of Didymos and Dimorphos, respectively. On the basis of the pre-impact orbit period and semimajor axis[6], and the volume-equivalent diameters provided in this work, we calculated a nominal system bulk density of $2{,}400 \pm 250$ kg m$^{-3}$ using the uncertainties quoted in the previous sentence. However, to represent additional possible systematic uncertainties, we adopted a slightly larger uncertainty, which gives a system bulk density of $2{,}400 \pm 300$ kg m$^{-3}$.

The porosity of Dimorphos was estimated as follows:

$$\phi = 1 - (\rho_{bulk}/\rho_{grain})$$

In this work, we assumed that Dimorphos's bulk density matches the entire system's bulk density. There are three near-Earth asteroid systems for which the satellite's bulk density has been independently measured: 66391 Moshup, 2000 DP107 and 2001 SN263 (two satellites). Two of those satellites (Squannit and 2001 SN263 gamma) were measured to be denser than the primary and two were less dense (2001 SN263 beta and 2000 DP107 beta)[20–22]. For all three systems, the $1\sigma$ uncertainties for the satellite and the primary densities overlap. Furthermore, recent work[49] estimated that the size of Squannit is about 30% larger than estimated previously[20], meaning that the bulk density of Squannit may be in better agreement with its primary. Given these other examples, assuming Dimorphos's density matches the system bulk density is a reasonable starting point, although its true density could differ substantially from this value. Hera will determine the masses and densities of Didymos and Dimorphos and test the validity of this assumption[39].

## Data availability

The DRACO images shown in this paper, the global digital terrain model of Dimorphos and the local digital terrain model of the impact site are available in a permanent archive associated with this paper in the JHU/APL Data Archive (https://lib.jhuapl.edu/papers/dart-an-autonomous-kinetic-impact-into-a-near-eart/). All raw and calibrated DRACO images, as well as higher-order products such as digital terrain models, will ultimately be available via the Planetary Data System (PDS) (https://pds-smallbodies.astro.umd.edu/data_sb/missions/dart/index.shtml) by October 2023.

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

**Acknowledgements** We thank P. Boie, R. Harvey, M. Hill, A. Johnson, C. Kim, J. Kim, D. O'Shaughnessy, N. Osiander, G. Ottman, L. Rodovskiy, M. Rodriguez, A. Smith, K. Volland and all of the other people who made the DART impact possible; and M. Bruck Syal and K. Kumamoto for their feedback on the manuscript. This work made use of the Small Body Mapping Tool. This work was supported by the DART mission, NASA Contract No. 80MSFC20D0004. This work was supported by the Italian Space Agency (ASI) within the LICIACube project (ASI-INAF agreement AC n. 2019-31-HH.0). P.S. and P.P. were supported by the Grant Agency of the Czech Republic, grant 20-04431S. B.J.B. was funded by the NASA DART Participating Scientist Program #20-DARTPSP20-0007. S.C. acknowledges funding from the Crosby Distinguished Postdoctoral Fellowship Program of the Department of Earth, Atmospheric and Planetary Science, Massachusetts Institute of Technology. G.S.C. was funded by UK Science and Technology Facilities Council Grant ST/S000615/1. F.F. acknowledges funding from the Swiss National Science Foundation (SNSF) Ambizione grant No. 193346. M.J. and S.D.R. acknowledge support by the Swiss National Science Foundation (project number 200021_207359), and from the European Union's Horizon 2020 research and innovation programme under grant agreement no. 870377 (project NEO-MAPP). T.K. is supported by Academy of Finland project 335595 and by institutional support RVO 67985831 of the Institute of Geology of the Czech Academy of Sciences. P.M. acknowledges funding support from the European Union's Horizon 2020 research and innovation programme under grant agreement no. 870377 (project NEO-MAPP), the CNRS through the MITI interdisciplinary programmes, CNES and ESA. N.M. and C.Q.R. acknowledge funding support from the European Commission's Horizon 2020 research and innovation programme under grant agreement no. 870377 (NEO-MAPP project) and support from the Centre National d'Etudes Spatiales (CNES). J.O. has been funded by grant No. PID2021-125883NB-C22 by the Spanish Ministry of Science and Innovation/State Agency of Research MCIN/AEI/10.13039/501100011033 and by 'ERDF A way of making Europe'. S.R.S. acknowledges support from the NASA DART Participating Scientist Program, award no. 80NSSC22K0318. J.K.S. acknowledges support from NASA award 80NSSC21K1014. J.M.T.-R. acknowledges financial support from the project PID2021-128062NB-I00 funded by Spanish MCIN/AEI/10.13039/501100011033. P.B. acknowledges funding support from Europlanet/University of Edinburgh and Technical University of Kenya. Part of this research was carried out at the Jet Propulsion Laboratory, California Institute of Technology, under a contract with the National Aeronautics and Space Administration.

**Author contributions** R.T.D., C.M.E. and O.S.B. jointly led the writing of this paper, identification of the impact site and shape modelling of Dimorphos. R.T.D. is the Dimorphos shape modelling lead. C.M.E. is the DRACO instrument scientist. O.S.B. is the Proximity Operations Working Group lead. N.L.C., A.S.R. and A.F.C. lead the DART Investigation Team, contributed to writing and revision of this paper, and coordinated inputs across the DART Investigation Team. E.Y.A. contributed substantial revisions to the manuscript and was the DART mission systems engineer. H.F.A. wrote portions of the manuscript and played a key role in the analysis and interpretation of DART data. E.D.A., A.L.A., J.A.A., A.R.B., D.L.B., J.B., S.B., M.H.C., G.C., M.W.C., P.S.E., A.S.F., Z.J.F., S.H., M.A.H., P.A.H.-D., P.M.H., S.N.J., M.A.J., J.W.J., T.O.K., F.E.L., D.M.M., M.E.M., I.W.M., J.R.R., E.L.R., J.E.R., L. M. Rodriguez, L. M. Roufberg, B.P.R., C.A.S., M.P.S., B.N.S., C.E.S., E.J.S., E.A.S., Z.B.T., J.R.T., B.T.T., A.T.V., D.V., D.S.W. and K.A.W. played integral roles in the engineering team to ensure DART intercepted Dimorphos. R.W.G., M.A.J. and H.N. contributed to the effort to identify the impact site. R.W.G. also assisted with shape modelling of Dimorphos. E.E.P. led the efforts to use DRACO images to determine the shape of Didymos, with inputs from O.S.B. and R.W.G. B.J.B. calculated the geometric albedo of Didymos. T.L.F. contributed to the calibration of DRACO and shape modelling efforts. D.S.W. led determination of the time of impact and wrote that part of the methods section. R.C.E., H.N. and C.D.W. developed the software used to process DRACO images and digital terrain models into data products. S.-Y.W.H. contributed to DRACO calibration. A.L., N.M., M.P., S.D.R., C.Q.R., A.M.S., D.M.G., M.E.D. and J.M.S. contributed to impact site characterization. T.K., H.F.A., M.C.N., D.J.S., J.M.T.-R. and Y.Z. contributed to the calculation and interpretation of the density of Dimorphos. S.R.C., N.A.M., S.P.N., C.A.T., P.P. and P.S. provided orbit solutions used to compute the GM of the Didymos system. E.I.A., P.B., R.-L.B., S.C., G.S.C., F.F., D.M.G., I.H. M.H., S.A.J., M.J., R.L., P.M., S.M., M.C.N., J.O., K.T.R., D.J.S., S.R.S and J.K.S. provided comments that substantively revised the manuscript. This work made use of the Small Body Mapping Tool (SBMT; sbmt.jhuapl.edu). R.J.S., J.M.P. and N.R.L. are the software developers for the SBMT. C.M.E., O.S.B., R.T.D. and A.C.M. also work on the SBMT team.

**Competing interests** The authors declare no competing interests.

**Additional information**
**Correspondence and requests for materials** should be addressed to R. Terik Daly.

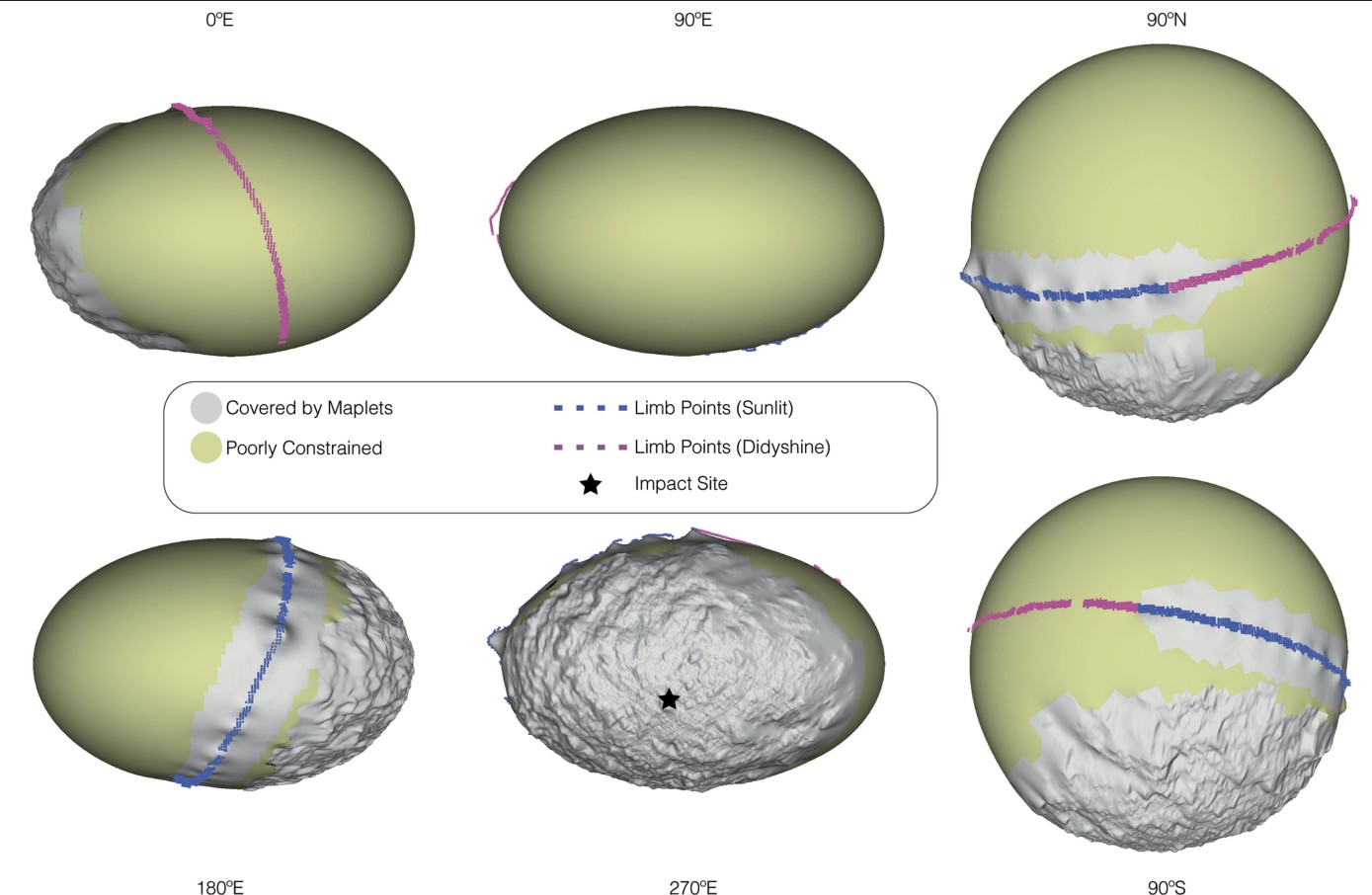

**Extended Data Fig. 1 | The Dimorphos global digital terrain model (DTM) as viewed along its principal axes.** The black star marks the DART impact site. Colors on the DTM indicate the various constraints used to build the model.

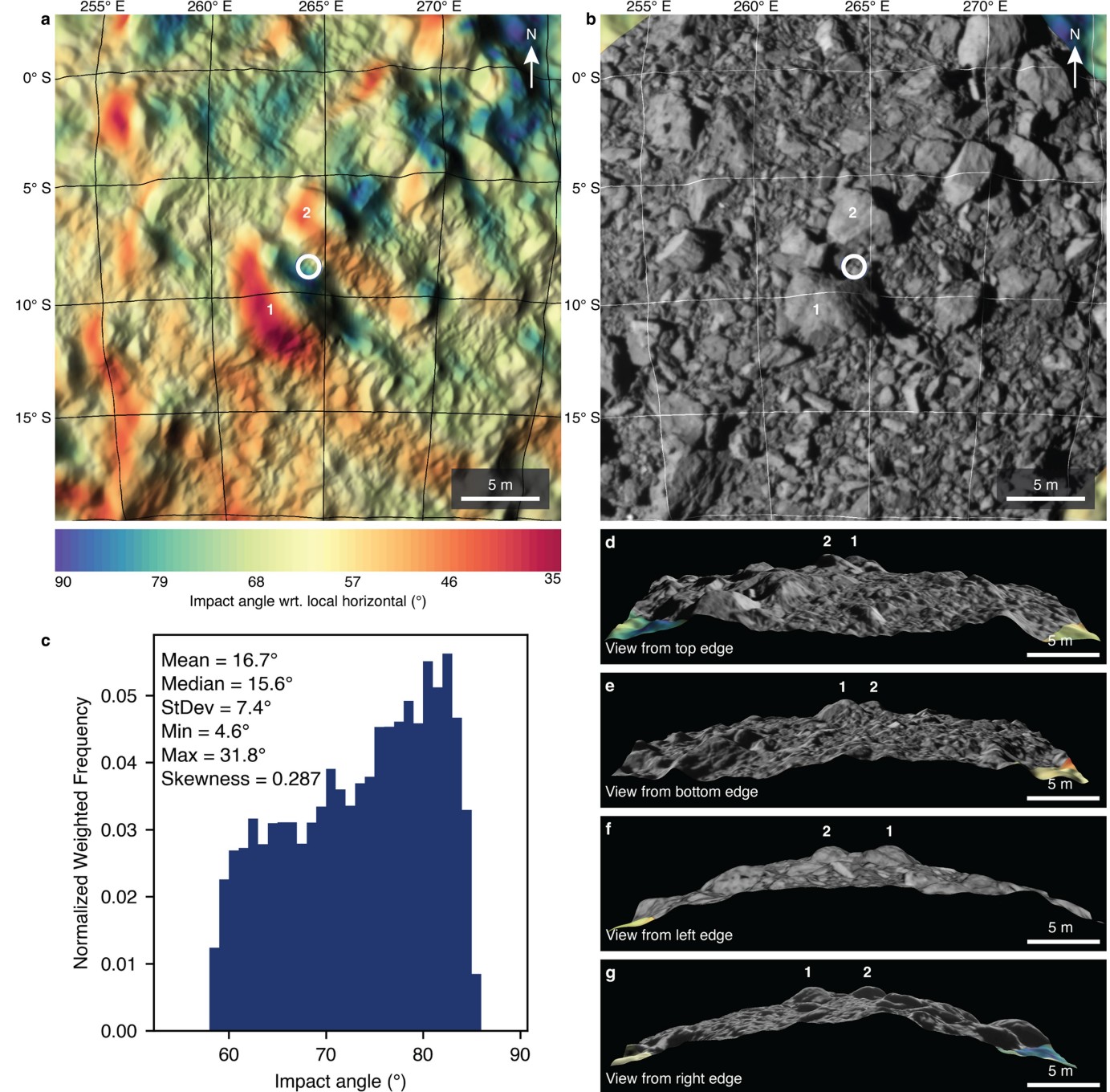

**Extended Data Fig. 2 | Tilts and topography at the impact site.** DTM of the DART impact site with facets colored by the impact angle with respect to local horizontal, averaged over a 3-m region. The DTM is lit to match the lightning in DRACO images at the time of impact. The white circle shows the uncertainty in the impact location (a circle with a radius of 68 cm). Boulders 1 and 2 correspond to boulders 1 and 2 in Fig. 2. (b) The same DTM with DRACO image dart_0401930048_45552_01_iof.fits draped over it. The image does not cover the entire DTM, so the corners of panel b show the impact angle plate coloring. (c) Histogram of tilts within the white circle representing the uncertainty in the impact site location. (d) – (g) Perspective views of the impact site DTM with overlaid image shown in (b), i.e., the DTM in panel (b) viewed edge on from each of the four sides of the DTM. Boulders 1 and 2 are prominent, as is the small niche between them in which the spacecraft bus hit the surface.

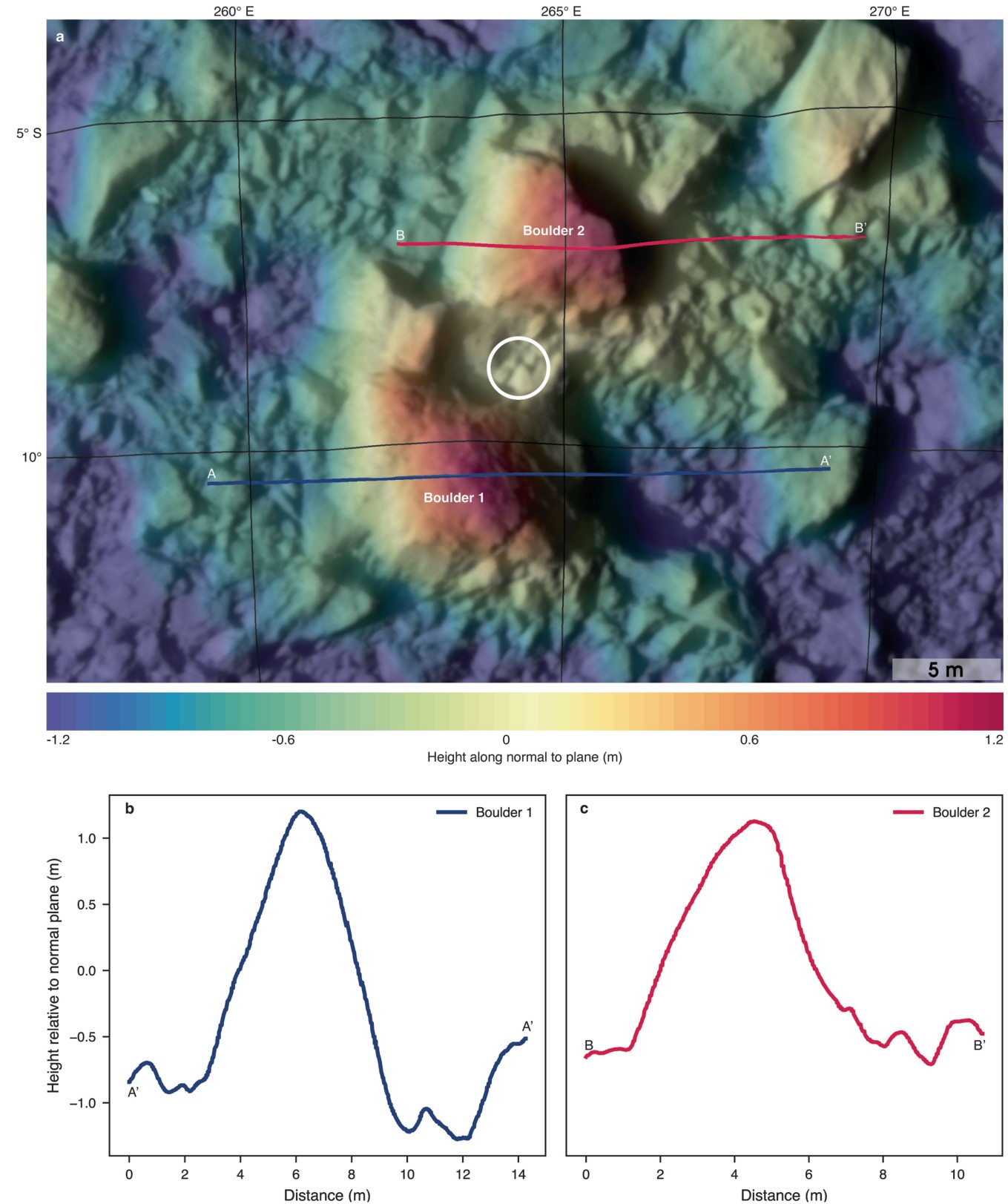

**Extended Data Fig. 3 | Boulders at the impact site.** (a) Zoomed-in view of the impact site DTM to focus on the two largest boulders near the impact site. Facets in the DTM are colored by the height of the facet along a normal to a plane fit to all of the points in the DTM. DRACO image dart_0401930048_45552_01_iof.fits is draped over the DTM at 40% opacity. The DTM is lit to match the lightning in the DRACO image. The white circle shows the uncertainty in the impact location (a circle with a radius of 68 cm). The red and blue paths show the locations of two topographic profiles across (b) boulder 1 from A to A' and (c) boulder 2 from B to B'.

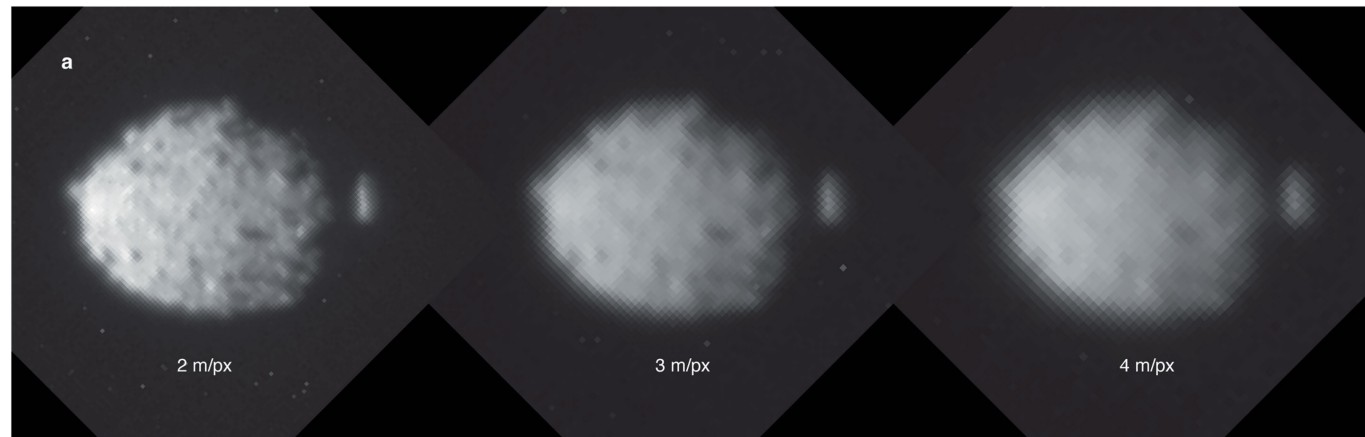

a

2 m/px 3 m/px 4 m/px

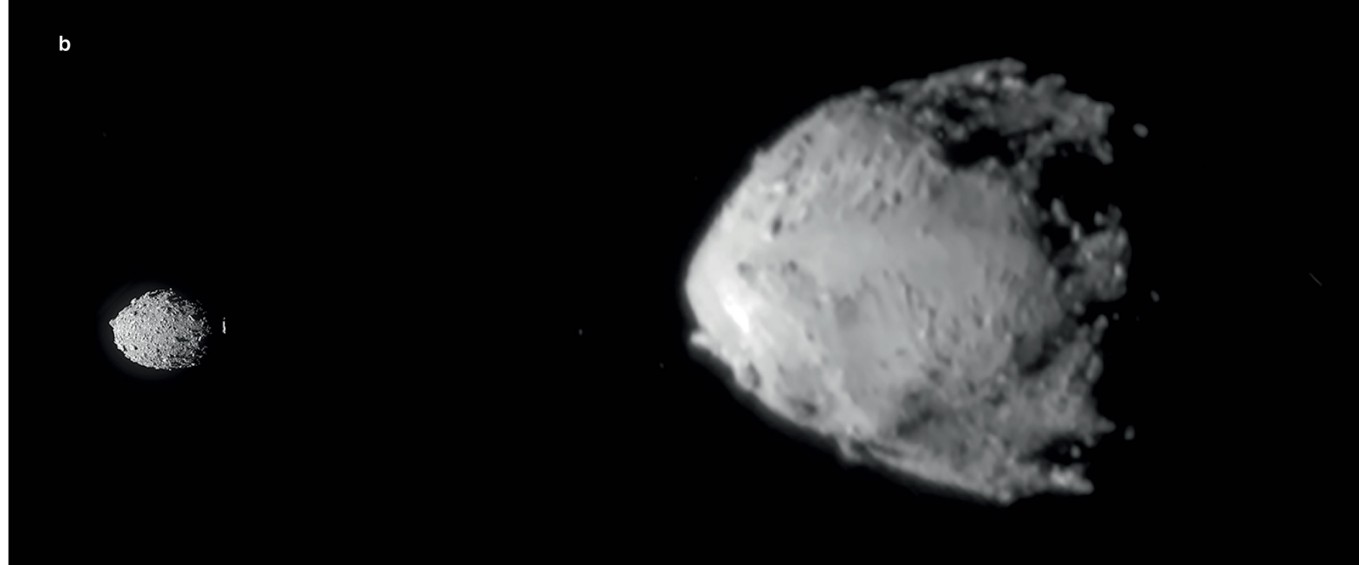

b

**Extended Data Fig. 4 | Dimorphos and Didymos as seen by DART.** (a) The asteroid Dimorphos seen at a range of pixel scales. Numerous boulders can be distinguished across the surface in images as coarse as 2–3 m pixel scale. Without the context of higher-resolution images, it would be difficult to definitively identify boulders in the 4-m pixel scale image. Image names (from left to right): dart_0401929985_18096_01_iof.fits, dart_0401929952_31226_01_iof.fits, dart_0401929919_44355_01_iof.fits. (b) Composite image of asteroids Dimorphos and Didymos. Dimorphos is at left; Didymos is at right. The two asteroids and the distance between them are to scale. This image was produced by combining two DRACO images to show Dimorphos at higher resolution than Didymos. In spite of the different resolutions, the two surfaces give different first impressions. Dimorphos has a boulder-rich surface with an ellipsoidal shape. Didymos exhibits boulders but also smoother areas and larger concavities. The north poles of Dimorphos and Didymos point to the top of the figure.

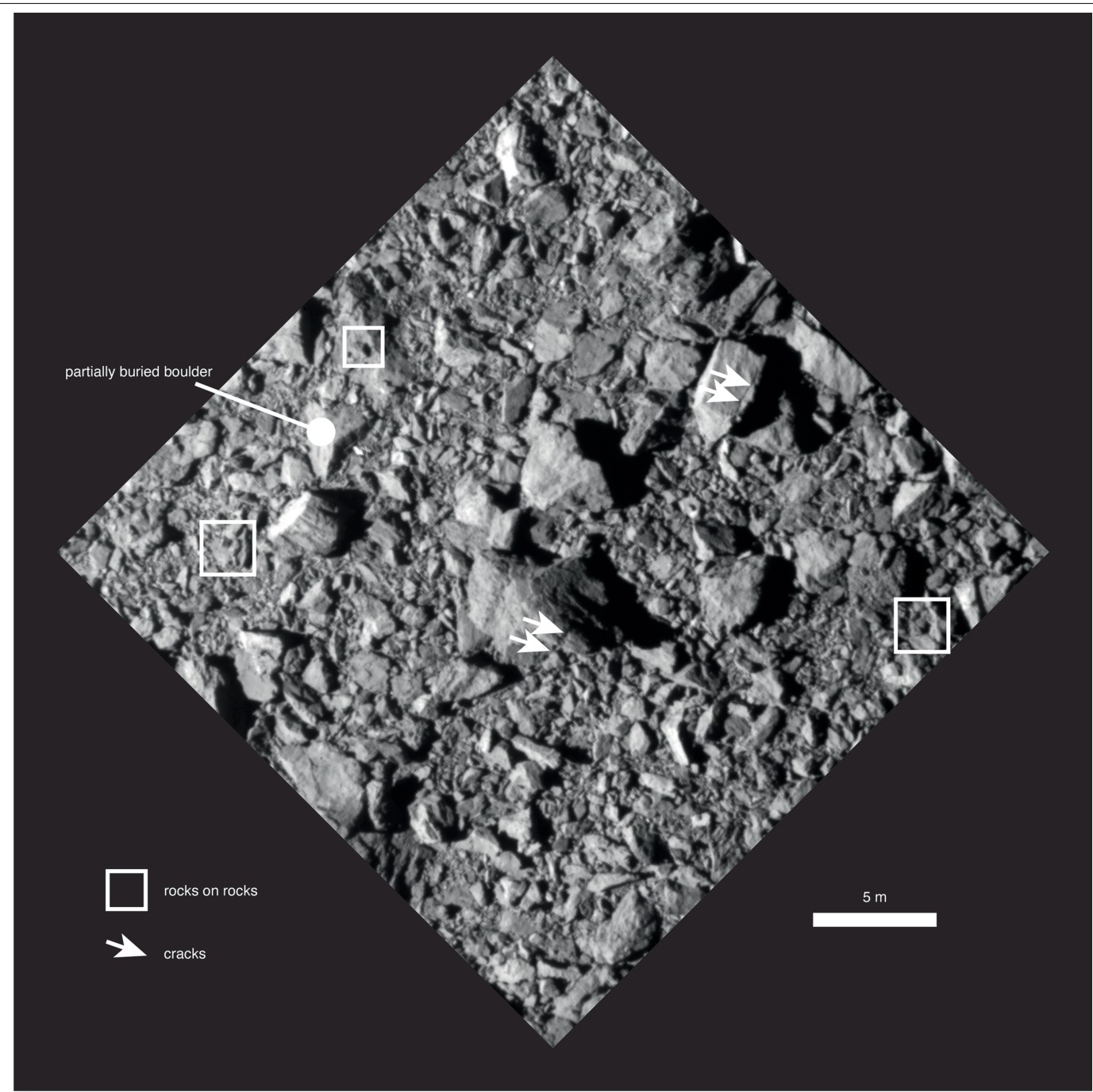

partially buried boulder

rocks on rocks

cracks

5 m

**Extended Data Fig. 5 | The final full DRACO image of Dimorphos's surface.** Examples of cracks (white arrows), rocks on rocks (squares) and a partially buried boulder are indicated. Image name: dart_0401930049_43695_01_iof.fits. The north pole of Dimorphos is toward the top of the figure.

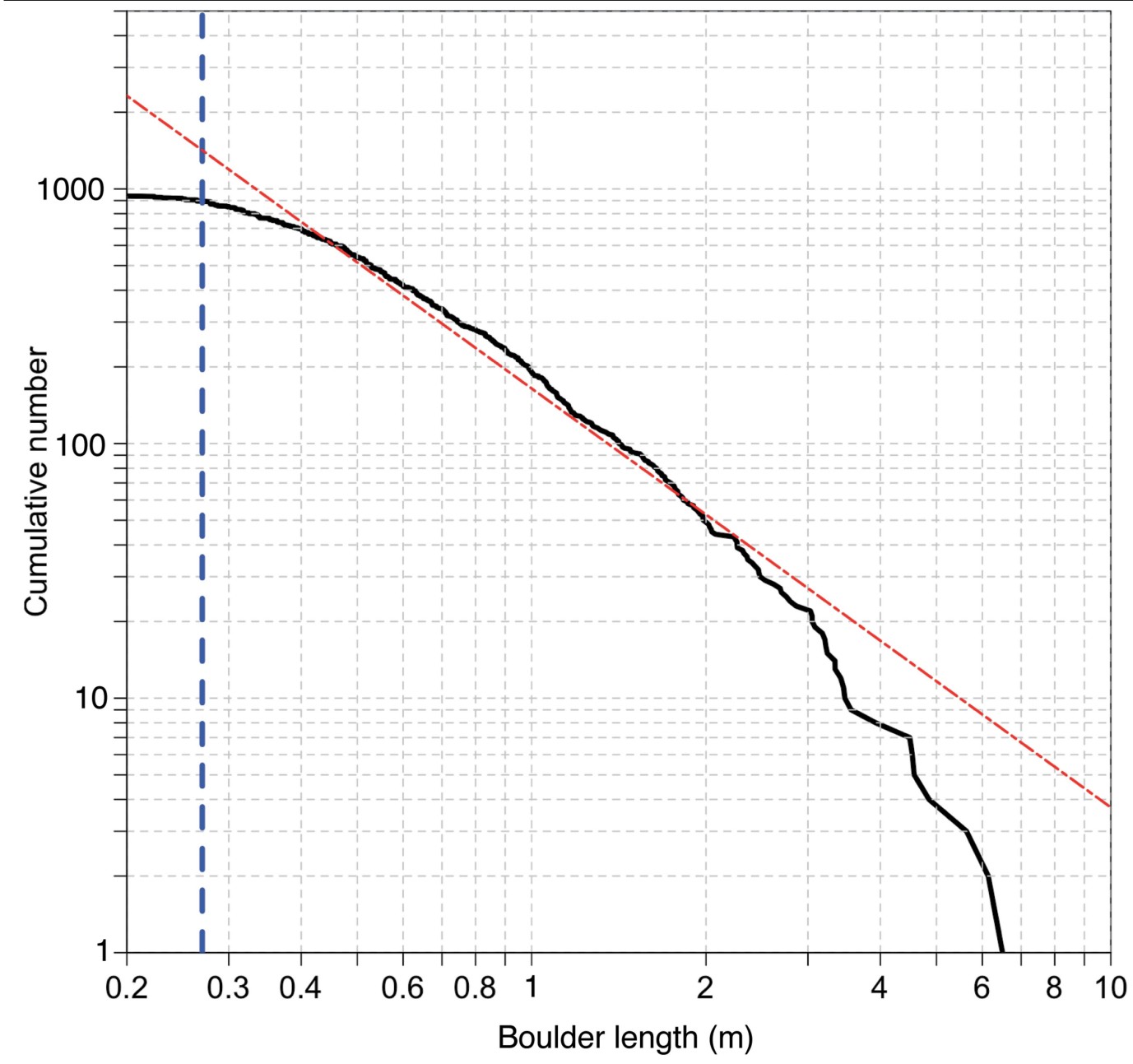

**Extended Data Fig. 6 | The size-frequency distribution of boulders identified in the last DRACO full image (dart_0401930049_43695_01_iof.fits).** The limit of image resolution, assuming ≥3 pixel sampling, is ~16.5 cm, so the overturn at small sizes is real and not an observational bias (Pajola et al. 2015). Here, a conservative 5-pixel sampling limit (27.5 cm) is indicated by the vertical blue dashed line. The distribution is not well described by a single power law (shown here as a red dot-dashed line with a slope of −1.65).

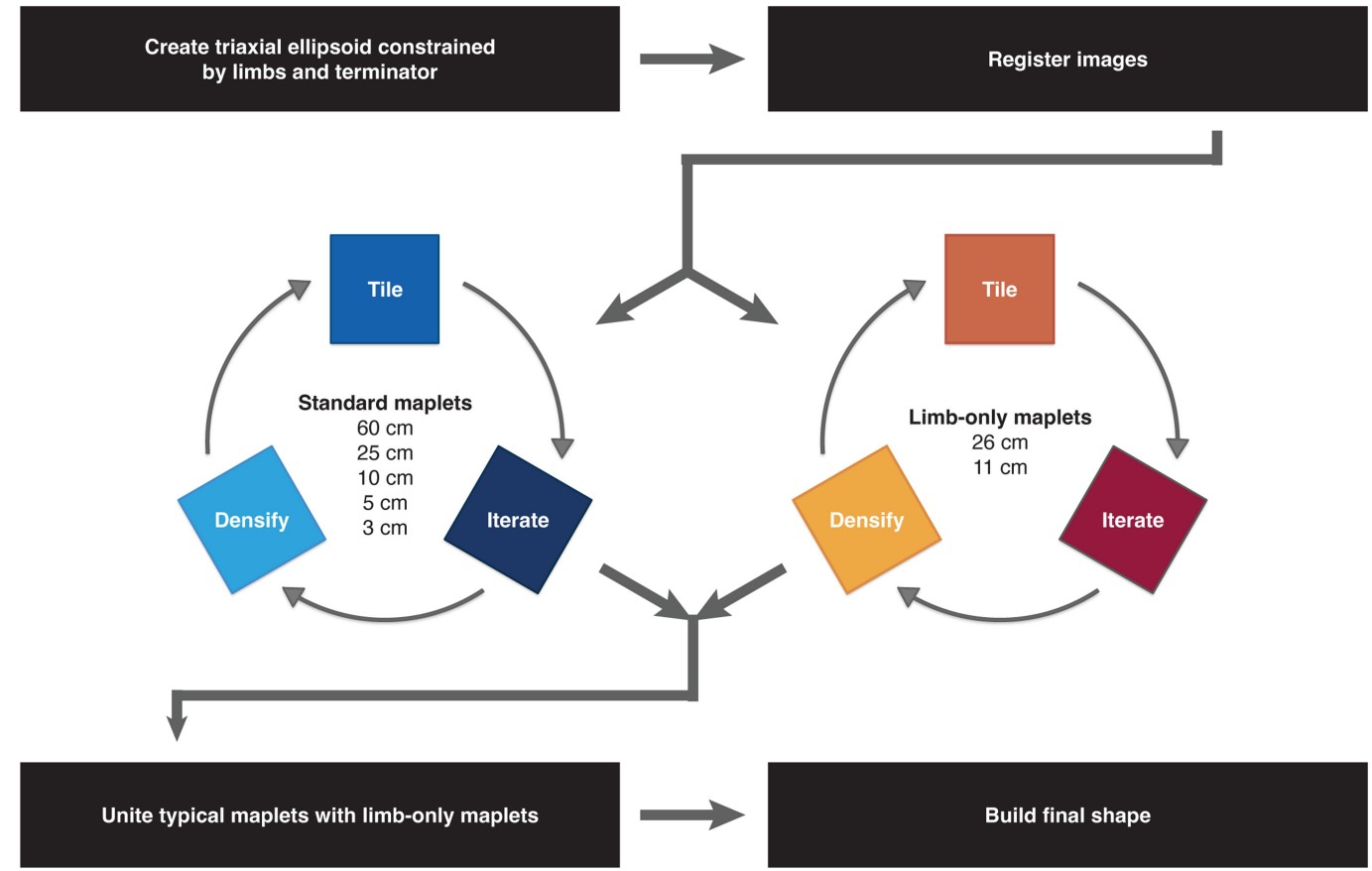

**Extended Data Fig. 7 | Shape modeling process used to build a global digital terrain model of Dimorphos.** The process was informed by shape modeling tests conducted prior to impact.

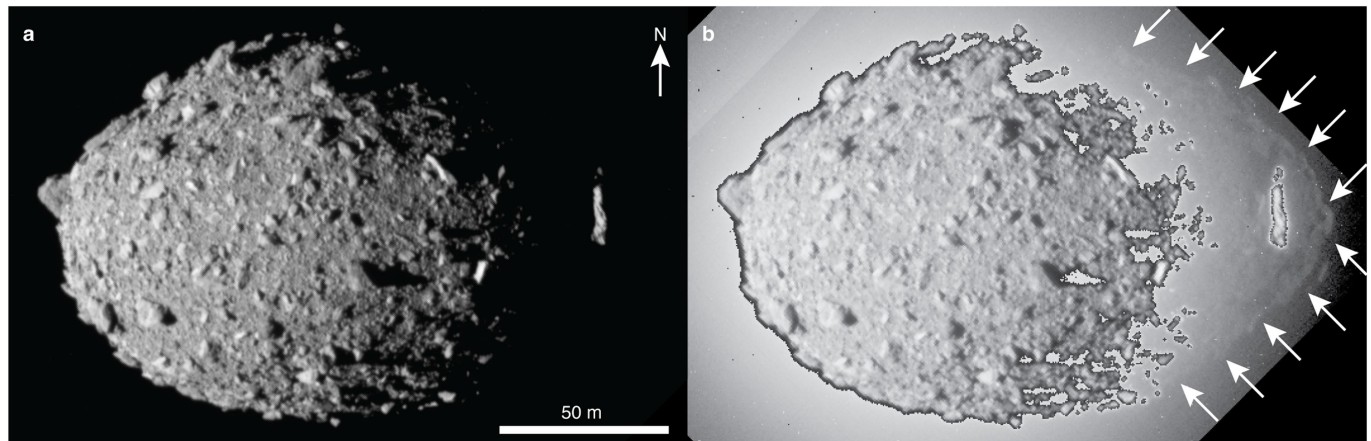

**Extended Data Fig. 8 | Sunlit and Didymos-lit limbs of Dimorphos.** The same image of Dimorphos stretched to optimize (a) the limb lit by the Sun and (b) the limb lit by light reflected off Didymos. In (b), pixels with I/F < 0.014 have been scaled up by a factor of 6 to allow the faint features to be seen along with the sunlit features, for better comparison to (a). Together, the two limbs reveal a complete outline of the asteroid as seen by DRACO. The image is dart_0401930039_14119_01_iof.fits.

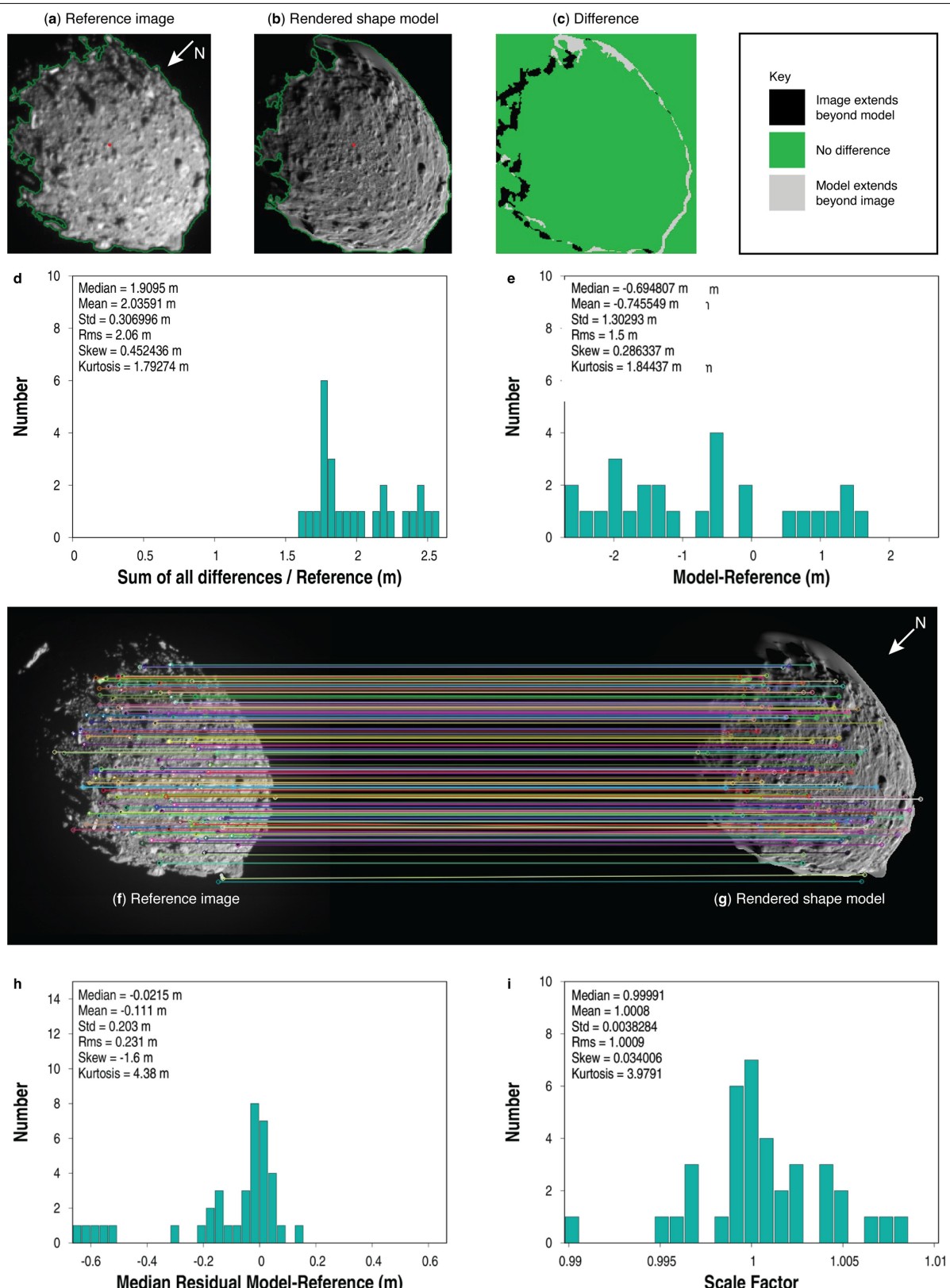

**(a)** Reference image

**(b)** Rendered shape model

**(c)** Difference

Key

Image extends beyond model

No difference

Model extends beyond image

**d**

Median = 1.9095 m
Mean = 2.03591 m
Std = 0.306996 m
Rms = 2.06 m
Skew = 0.452436 m
Kurtosis = 1.79274 m

Number

Sum of all differences / Reference (m)

**e**

Median = −0.694807 m      m
Mean = −0.745549 m      n
Std = 1.30293 m
Rms = 1.5 m
Skew = 0.286337 m
Kurtosis = 1.84437 m      n

Number

Model-Reference (m)

**(f)** Reference image

**(g)** Rendered shape model

**h**

Median = −0.0215 m
Mean = −0.111 m
Std = 0.203 m
Rms = 0.231 m
Skew = −1.6 m
Kurtosis = 4.38 m

Number

Median Residual Model-Reference (m)

**i**

Median = 0.99991
Mean = 1.0008
Std = 0.0038284
Rms = 1.0009
Skew = 0.034006
Kurtosis = 3.9791

Number

Scale Factor

**Extended Data Fig. 9** | See next page for caption.

**Extended Data Fig. 9 | Shape model assessments.** (a)–(e) show results from limb/terminator shape model assessments. Panels (a)–(c) show an example of (a) a reference DRACO image, (b) the rendered shape model with the same lighting and viewing geometry as the reference image, and (c) the difference between the model and the reference image. (d) and (e) show results from limb/terminator assessments from many DRACO images. (d) Sum of the absolute value of the image-model differences, normalized by the image perimeter. The median is the most relevant measure of uncertainty from this metric because the distribution is always one-sided and never gaussian. (e) The differences in the radii of the equivalent-area circles for the reference image and rendered shape model, respectively. The radius of the equivalent-area circle is the radius of a circle with the same area as the total area of lit terrain on either the rendered model or the reference image. The mean is the most relevant measure of uncertainty from this metric because the distribution should be symmetric. (f)–(i) show results from keypoint assessments. The colored lines in panels (f) and (g) connect features matched by the algorithm in the image and on the shape model. Most, but not all, matches are reasonable, so the median value based on all keypoints is used. (h) shows a metric derived from differences between corresponding keypoints across several tens of DRACO images. (i) shows a model-to-image scale factor derived by comparing the distances measured between all keypoints in the reference DRACO image to the distances measured between all keypoints in images of the rendered shape model. The arrows in panels (a) and (g) indicate the direction of Dimorphos north (+Z).

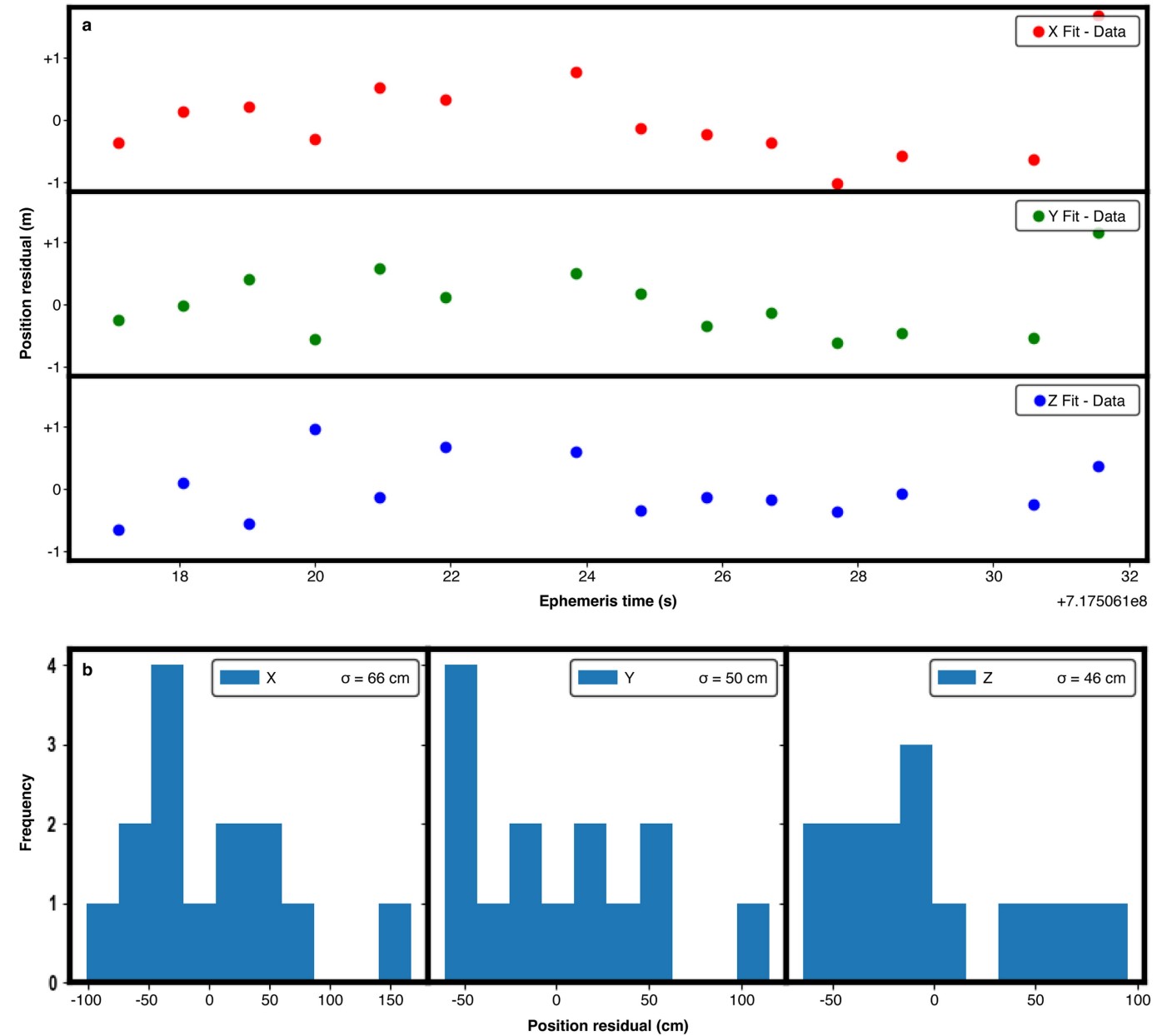

**Extended Data Fig. 10 | Fit residuals.** (a) Spacecraft position residuals and (b) residual distributions estimated after fitting the spacecraft locations obtained during the SPC shape modeling process with a second-order polynomial all in the J2000 inertial frame relative to Dimorphos. Uncertainties attributed to the location of the impact site and reported in Table 1 are in the Y and Z axes.