## [Peer Review File · Nature]

Manuscript Title: Successful Kinetic Impact into an Asteroid for Planetary Defense

Reviewer Comments & Author Rebuttals

Reviewer Reports on the Initial Version:

Referees' comments:

Referee #1 (Remarks to the Author):

This article describes the reconstructed timing, states, and geometries associated with the autonomous DART kinetic impacting event on Didymos's moon, Dimorphos. The authors describe the methodology by which they reduced the final estimates of the properties of the DART impact, as well as geophysical properties of both Dimorphos and Didymos from the limited mission data. This work appropriately cites recent, relevant data reduction methods (e.g. OSIRIS-REx) to the novel DART mission dataset, which provide the community with a quality reconstruction of the impact location, angle, speed, as well estimates of as each object's dimensions, volume, mass, and density. Due to data and geometry challenges, formal statistical uncertainties can under-represent the true uncertainty in the estimate, and an uncertainty value must be scaled or derived by alternate means or philosophies. In my opinion, the authors appropriately describe in the methodology section the rationale for their chosen uncertainty values, which include a reasonable level of conservatism. The reporting of the data and methodology is sufficiently detailed and transparent. The abstract gives great context for the work, but feels too introductory and lacking in a description of what DART mission results are included. This work focuses primarily on the event reconstruction, which should be mentioned; especially given the vagueness of the title, and lack of discussion related to the orbit dynamics change [6] outside of the abstract. Overall, this is a well written paper and I look forward to seeing it published.

Editorial: Line 336 misspells 'stabilized'.

Transparency FTW,
Coralie Adam

Referee #2 (Remarks to the Author):

The manuscript summarizes the preliminary results of the DART mission, demonstrating an autonomous kinetic impact experiment into the secondary asteroid Dimorphos, in the Didymos binary system. The manuscript describes the mission objectives and timeline, and the mission results in determining the size and shape of Dimorphos (and Didymos), and in characterizing the surface and impact point on Dimorphos' surface. The DART mission represents a major advancement in our capability to successfully implement kinetic impacts on near-Earth asteroids, and this manuscript will serve as a key reference in advancing binary asteroid science. The manuscript is clearly written with

supportive figures and explanations, and appropriate methodology with sound and well-justified assumptions. I have relatively minor comments which are summarized below.

General Comments:

1. Determining the impact site. The impact site was determined based on the trajectory of the spacecraft bus, extrapolated down to the surface. However, the spacecraft also contains two 8.5 m solar arrays in addition to the bus. Based on the images and DTM of the impact site, and the incoming trajectory of the spacecraft (Fig. 3), it appears that the solar arrays would impact the elevated portions of boulder 1 (and potentially boulder 2) before the smaller spacecraft bus could impact the surface at the stated region (between boulders 1 and 2, Figure 2). Did the propagation of the velocity vector onto the surface (e.g., Line 454-455) consider roughness of the surface generated by large boulders or the irregular spacecraft shape? I could not find the height of boulders 1 and 2 shown in the figures or in the text—it would be helpful to include a local DTM showing facet radius, similar to Extended Data Figure 2 showing impact angle. Also, how do boulder 1 and 2 heights compare to the ~1 m position residuals shown in Extended Data Figure 10? The presence of the solar arrays and their impact on the determination of the impact site should be discussed in the manuscript.

2. Broader results from the mission. The resulting crater that formed from the impact is not discussed, nor is the change to the orbit of the Dimorphos system. What is the predicted size (and characteristics) of the crater formed from the impact? For example, given the blocky nature and small facet size of the DTM of the impact site, is impact armoring anticipated to have operated? Additionally, a preliminary prediction of the change to the system orbital period would strengthen the broader impact of the manuscript.

Specific Comments:

- Line 87-95: The new estimate of Dimorphos' diameter from DART results is not included in this text, only the previous estimate from ground-based radar observations (150 m). Consider providing it in the text here as well as in Table 1 for clarity. Didymos' size is also not specifically listed here, only in Table 1.

- Figure 2 and figure caption: A couple of clarifying points for the caption and figure. Part (a) the spacecraft footprint in the white dashed box is very small and thin, so is barely visible in the figure. Consider adding an arrow or label, and specifying in the text that the spacecraft is the thin vertical line in the white dashed box. This white dashed box in (a) is different from the extent of the white dashed box in (b), which shows the extent of part (c). I'd recommend making the box solid in (a) and stating this in the figure caption. Also in Line 121, "...with a white square indicating..." should specify "solid white square" to distinguish it from the dashed line.

- Line 155-156 "The longest axes of boulders in the final complete image are 0.16 m to 6.5 m in length": This statement implies that every particle in the final image is resolvable and larger than 16 cm diameter—is this the case, that there is no proportion of unresolved material in the site? This would be of interest to the community to compare against the surfaces and block size frequency distributions of other NEAs such as Itokawa, Bennu, and Ryugu.

- Line 374/377: Specific figure part callouts would help clarify the text here: Line 374 "Extended Data Fig. 9a-e" and Line 377 "Extended Data Fig. 9f-i"

- Line 387 "...indicate that the asteroid is...": Presumably "asteroid" here refers to the asteroid shape model, correct? If so, this could be clarified.

- Line 416-42: These few sentences would more naturally follow from the discussion of X and Y uncertainty, before the discussion of volume. Consider moving this text up to follow Line 395.

- Line 465-466 "Therefore, we calculated a mean tilt with respect to the impact velocity vector for each facet in the impact site DTM.": Is the mean tilt over the impact site the most characteristic impact angle? If there are higher-tilted facets present (even if they do not dominate the site), wouldn't they interact with the spacecraft bus and alter the effective impact angle?

- Line 550-551: The provided link to the JHU/APL Data Archive did not work (all other links were functional)

- Extended Data Fig. 1: It would be useful to have the impact site marked on the global DTM on this figure.

- Extended Data Fig. 9: (d) why does the x-axis include negative values if the parameter is the sum of absolute value differences? (e) the best-fit circle referenced here is not discussed in the text, please expand on the description either here or in the text.

- Extended Data Figure 10: why is position residual expressed in terms of km? Readability would be improved if it were in terms of meters.

- Please do a final proof-read of the text, I found a few stray spaces throughout (e.g., Line 430)

Author Rebuttals to Initial Comments:

We have reviewed the editorial and reviewer comments and revised the manuscript accordingly. Below, we provide a point-by-point response to the reviewer comments. Reviewer comments are italicized; my response is indented and not italicized. Revised text is colored **bold blue** in the manuscript. Text that was moved in the main text or methods (but not otherwise revised) is shown in *italics*.

Referee #1 (Remarks to the Author):

This article describes the reconstructed timing, states, and geometries associated with the autonomous DART kinetic impacting event on Didymos's moon, Dimorphos. The authors describe the methodology by which they reduced the final estimates of the properties of the DART impact, as well as geophysical properties of both Dimorphos and Didymos from the limited mission data. This work appropriately cites recent, relevant data reduction methods (e.g. OSIRIS-REx) to the novel DART mission dataset, which provide the community with a quality reconstruction of the impact location, angle, speed, as well estimates of as each object's dimensions, volume, mass, and density. Due to data and geometry challenges, formal statistical uncertainties can under-represent the true uncertainty in the estimate, and an uncertainty value must be scaled or derived by alternate means or philosophies. In my opinion, the authors appropriately describe in the methodology section the rationale for their chosen uncertainty values, which include a reasonable level of conservatism. The reporting of the data and methodology is sufficiently detailed and transparent. The abstract gives great context for the work, but feels too introductory and lacking in a description of what DART mission results are included. This work focuses primarily on the event reconstruction, which should be mentioned; especially given the vagueness of the title, and lack of discussion related to the orbit dynamics change [6] outside of the abstract. Overall, this is a well written paper and I look forward to seeing it published.

Editorial: Line 336 misspells 'stabilized'.

*Transparency FTW,
Coralie Adam*

We revised the “Here we...” sentence in the first paragraph to state that this paper reconstructs the timeline leading to impact, the location and nature of the impact site, and the size and shape of Dimorphos.

In terms of the overall scope, note that this paper is one of a set of papers being submitted to *Nature* about the DART mission. This paper focuses on the autonomous impact and the impact site. A second paper focuses on the period change. A third paper focuses on the ejecta as observed by the Hubble Space Telescope. A fourth paper focuses on determining the momentum enhancement factor for the DART impact. Per guidance from the editor, we expanded the first paragraph of this paper slightly to provide additional context for all of the DART papers.

We corrected the typo.

Referee #2 (Remarks to the Author):

The manuscript summarizes the preliminary results of the DART mission, demonstrating an autonomous kinetic impact experiment into the secondary asteroid Dimorphos, in the Didymos binary system. The manuscript describes the mission objectives and timeline, and the mission results in determining the size and shape of Dimorphos (and Didymos), and in characterizing the surface and impact point on Dimorphos' surface. The DART mission represents a major advancement in our capability to successfully implement kinetic impacts on near-Earth asteroids, and this manuscript will serve as a key reference in advancing binary asteroid science. The manuscript is clearly written with supportive figures and explanations, and appropriate methodology with sound and well-justified assumptions. I have relatively minor comments which are summarized below.

General Comments:

1. Determining the impact site. The impact site was determined based on the trajectory of the spacecraft bus, extrapolated down to the surface. However, the spacecraft also contains two 8.5 m solar arrays in addition to the bus. Based on the images and DTM of the impact site, and the incoming trajectory of the spacecraft (Fig. 3), it appears that the solar arrays would impact the elevated portions of boulder 1 (and potentially boulder 2) before the smaller spacecraft bus could impact the surface at the stated region (between boulders 1 and 2, Figure 2). Did the propagation of the velocity vector onto the surface (e.g., Line 454-455) consider roughness of the surface generated by large boulders or the irregular spacecraft shape? I could not find the height of boulders 1 and 2 shown in the figures or in the text—it would be helpful to include a local DTM showing facet radius, similar to Extended Data Figure 2 showing impact angle. Also, how do boulder 1 and 2 heights compare to the ~1 m position residuals shown in Extended Data Figure 10? The presence of the solar arrays and their impact on the determination of the impact site should be discussed in the manuscript.

We expanded the main text to discuss the solar arrays and their role in the DART impact. The impact site reconstruction was done using the entire spacecraft in the DART spacecraft frame, not only with the bus. We updated Figure 2 to show the outlines of the solar panels, in addition to the outline of the spacecraft bus. This addition clarifies relationships between the two large boulders and the solar arrays. We also added sentences to the main text to explain the relative timing of contact of the solar panels and bus with the asteroid. We updated Figure 3 to illustrate these points more clearly.

The impact site assessment was done on the full 3-million plate global shape model of Dimorphos, so the propagation of the reconstructed spacecraft trajectory does account for the topography of the impact site. In addition to propagating the reconstructed trajectory to the point of intersection, we also calculated the spacecraft states at a few different

heights above the shape model to clarify the order in which the solar arrays and spacecraft bus contacted Dimorphos. We added a sentence to the “impact site identification” section of the methods to clarify this point.

We combined Extended Data Figs. 3 and 4 into a single figure, Extended Data Fig. 4, to make room for the reviewer’s requested figure showing the heights of the two boulders near the impact site. This new figure is Extended Data Figure 3 in the revised manuscript. Instead of showing facet radius, however, we show height normal to a plane fit to all the points in the impact site DTM. This approach shows local height differences more clearly than facet radius, which is dominated by the curvature of the asteroid. To measure the boulder heights, we extracted profiles from the impact site DTM across the highest point of each boulder. The boulder height is the difference between the tallest point on the boulder and the background terrain on either side of the boulder in the impact site DTM. The estimated heights of the boulders from the impact site DTM are ~2.2 m for boulder 1 and ~1.6 m for boulder 2, heights that exceed the position residuals shown in Extended Data Figure 10.

The boulder heights may be underestimates because SPC tends to smooth boulders, even when ideal imaging data exists (Barnouin et al., 2020, Digital Terrain Mapping by the OSIRIS-REx mission, *PSS*, <https://doi.org/10.1016/j.pss.2019.104764>). As noted on lines 390 – 394 of the original manuscript, assessments of shadow heights indicate that some boulders are well represented in the topography but others are somewhat too short. Shadow lengths suggest that height of boulder 1 in the impact site DTM is ~10% too small, but the height of boulder 2 in the impact site DTM is correct. The errors in boulder heights derived from shadows are much smaller than the spacecraft position residuals.

2. Broader results from the mission. The resulting crater that formed from the impact is not discussed, nor is the change to the orbit of the Dimorphos system. What is the predicted size (and characteristics) of the crater formed from the impact? For example, given the blocky nature and small facet size of the DTM of the impact site, is impact armoring anticipated to have operated? Additionally, a preliminary prediction of the change to the system orbital period would strengthen the broader impact of the manuscript.

As noted in the response to the first reviewer’s comments, this paper is part of a set of papers being submitted to *Nature*. This paper focuses on the autonomous DART impact, the impact site, and shape of Dimorphos. A second paper focuses on the period change. A third paper focuses on the ejecta as observed by the Hubble Space Telescope. A fourth paper focuses on determining the momentum enhancement factor for the DART impact. Other papers are being written to discuss ranges for the possible size and properties of the crater formed by the DART impact, but that significant effort is not within the scope of this paper. This first ensemble of papers will paint the broader results from the mission, and this specific paper is intended to set the stage for the other work.

Specific Comments:

- *Line 87-95: The new estimate of Dimorphos' diameter from DART results is not included in this text, only the previous estimate from ground-based radar observations (150 m). Consider providing it in the text here as well as in Table 1 for clarity. Didymos' size is also not specifically listed here, only in Table 1.*

We added the diameter for the volume-equivalent sphere to the main text. However, since we left Didymos in the table only because the focus of the bulk of this paper is on Dimorphos. (The shape of Didymos factors only into the total volume of the system and therefore the reported density.) A separate paper is being prepared that focuses on the shape of Didymos.

- *Figure 2 and figure caption: A couple of clarifying points for the caption and figure. Part (a) the spacecraft footprint in the white dashed box is very small and thin, so is barely visible in the figure. Consider adding an arrow or label, and specifying in the text that the spacecraft is the thin vertical line in the white dashed box. This white dashed box in (a) is different from the extent of the white dashed box in (b), which shows the extent of part (c). I'd recommend making the box solid in (a) and stating this in the figure caption. Also in Line 121, "...with a white square indicating..." should specify "solid white square" to distinguish it from the dashed line.*

We added an arrow to Figure 2a to mark the spacecraft. We made the square in 2a solid, instead of dashed, and updated the figure caption to reflect these changes.

- *Line 155-156 "The longest axes of boulders in the final complete image are 0.16 m to 6.5 m in length": This statement implies that every particle in the final image is resolvable and larger than 16 cm diameter—is this the case, that there is no proportion of unresolved material in the site? This would be of interest to the community to compare against the surfaces and block size frequency distributions of other NEAs such as Itokawa, Bennu, and Ryugu.*

We reworded this sentence to be more precise: the sizes of boulders that were counted to create the SFD ranged from 0.16 to 6.5 m. It is true that we see no evidence for extensive areas of smooth material, as stated later in the paragraph.

- *Line 374/377: Specific figure part callouts would help clarify the text here: Line 374 "Extended Data Fig. 9a-e" and Line 377 "Extended Data Fig. 9f-i"*

We added the suggested text.

- *Line 387 "...indicate that the asteroid is...": Presumably "asteroid" here refers to the asteroid shape model, correct? If so, this could be clarified.*

Yes, the asteroid shape model. We revised the text to clarify this point.

- Line 416-42: *These few sentences would more naturally follow from the discussion of X and Y uncertainty, before the discussion of volume. Consider moving this text up to follow Line 395.*

We moved this text as suggested.

- Line 465-466 *“Therefore, we calculated a mean tilt with respect to the impact velocity vector for each facet in the impact site DTM.”: Is the mean tilt over the impact site the most characteristic impact angle? If there are higher-tilted facets present (even if they do not dominate the site), wouldn't they interact with the spacecraft bus and alter the effective impact angle?*

The average mean tilt we report is weighted by facet area, so the mean tilt calculation does represent the characteristic impact angle. In addition, the mean tilt for any given facet explicitly includes information about the tilts of facets that surround it. The part of the spacecraft bus that contacted the surface had an area 1.56 m^2 , which means that it interacted with an area equivalent to ~ 1100 facets in the impact site DTM. (The average facet size in the impact site DTM near the impact site is 0.0014 m^2 . The minimum area of a facet in the impact site DTM is 0.0012 m^2 , and the maximum area of a facet is 0.0025 m^2 .) The fact the spacecraft bus contacted an area that is equivalent to so many facets in the shape model reduces the effect that any one facet would have.

- Line 550-551: *The provided link to the JHU/APL Data Archive did not work (all other links were functional)*

We checked the permissions on this link and believe that it is publicly accessible. We tested it on mobile devices and laptops. If there continue to be issues with this link, please let us know so that we can troubleshoot further.

- *Extended Data Fig. 1: It would be useful to have the impact site marked on the global DTM on this figure.*

We updated this figure to show the impact site.

- *Extended Data Fig. 9: (d) why does the x-axis include negative values if the parameter is the sum of absolute value differences? (e) the best-fit circle referenced here is not discussed in the text, please expand on the description either here or in the text.*

In regards to 9d, the fact that the x-axis included negative values was an oversight. We updated the plot so that the x-axis starts at 0.

In regards to 9e, we updated the figure caption to better describe this metric.

- *Extended Data Figure 10: why is position residual expressed in terms of km? Readability*

would be improved if it were in terms of meters.

We changed the units on the figure axis.

- Please do a final proof-read of the text, I found a few stray spaces throughout (e.g., Line 430)

We checked for duplicate spaces throughout and fixed a typo that the other reviewer caught.